# A multilayer circuit architecture for the generation of distinct locomotor behaviors in *Drosophila*

Aref Arzan Zarin[1†‡*], Brandon Mark[1†], Albert Cardona[2], Ashok Litwin-Kumar[3], Chris Q Doe[1*]

[1]Institute of Neuroscience, Howard Hughes Medical Institute, University of Oregon, Eugene, United States; [2]Janelia Research Campus, Howard Hughes Medical Institute, Ashburn, United States; [3]Mortimer B Zuckerman Mind Brain Behavior Institute, Department of Neuroscience, Columbia University, New York, United States

**Abstract** Animals generate diverse motor behaviors, yet how the same motor neurons (MNs) generate two distinct or antagonistic behaviors remains an open question. Here, we characterize *Drosophila* larval muscle activity patterns and premotor/motor circuits to understand how they generate forward and backward locomotion. We show that all body wall MNs are activated during both behaviors, but a subset of MNs change recruitment timing for each behavior. We used TEM to reconstruct a full segment of all 60 MNs and 236 premotor neurons (PMNs), including differentially-recruited MNs. Analysis of this comprehensive connectome identified PMN-MN 'labeled line' connectivity; PMN-MN combinatorial connectivity; asymmetric neuronal morphology; and PMN-MN circuit motifs that could all contribute to generating distinct behaviors. We generated a recurrent network model that reproduced the observed behaviors, and used functional optogenetics to validate selected model predictions. This PMN-MN connectome will provide a foundation for analyzing the full suite of larval behaviors.

*For correspondence:
azarin@bio.tamu.edu (AAZ);
cdoe@uoregon.edu (CQD)

†These authors contributed equally to this work

Present address: ‡Department of Biology, Texas A&M University, College Station, United States

Competing interests: The authors declare that no competing interests exist.

## Introduction

Locomotion is a rhythmic and flexible motor behavior that enables animals to explore and interact with their environment. Birds and insects fly, fish swim, limbed animals walk and run, and soft-body invertebrates crawl. In all cases, locomotion results from coordinated activity of muscles with different biomechanical outputs. This precisely regulated task is mediated by neural circuits composed of motor neurons (MNs), premotor interneurons (PMNs), proprioceptors, and descending command-like neurons (*Marder and Bucher, 2001*; *Arber, 2017*; *Arber and Costa, 2018*). A partial map of neurons and circuits regulating rhythmic locomotion have been made in mouse (*Crone et al., 2008*; *Grillner and Jessell, 2009*; *Zagoraiou et al., 2009*; *Dougherty et al., 2013*; *Goetz et al., 2015*; *Bikoff et al., 2016*), cat (*Kiehn, 2006*; *Nishimaru and Kakizaki, 2009*), fish (*Kimura et al., 2013*; *Song et al., 2016*), tadpole (*Roberts et al., 2008*; *Roberts et al., 2010*), lamprey (*Grillner, 2003*; *Mullins et al., 2011*), leech (*Brodfuehrer and Thorogood, 2001*; *Kristan et al., 2005*; *Marin-Burgin et al., 2008*; *Mullins et al., 2011*), crayfish (*Mulloney and Smarandache-Wellmann, 2012*; *Mulloney et al., 2014*), and worm (*Tsalik and Hobert, 2003*; *Wakabayashi et al., 2004*; *Haspel et al., 2010*; *Kawano et al., 2011*; *Piggott et al., 2011*; *Wen et al., 2012*; *Zhen and Samuel, 2015*; *Roberts et al., 2016*). These pioneering studies have provided a wealth of information on motor circuits, but with the exception of *C. elegans* (*White et al., 1986*), there has been no system where all MNs and PMNs have been identified and characterized. Thus, we are missing a

comprehensive picture of how an ensemble of interconnected neurons generates diverse locomotor behaviors.

Muscle recruitment patterns during different locomotor behaviors have been previously studied in multiple organisms, including human (*Grasso et al., 1998*; *van Deursen et al., 1998*; *Neptune et al., 2000*), cat (*Buford and Smith, 1990*), stick insect (*Gruhn et al., 2006*; *Tóth et al., 2012*), and leech (*Friesen and Kristan, 2007*). In the case of forward and backward locomotion, it has been suggested that only a subset of muscles change their activity timing in one behavior versus another, indicating an overall similarity in muscle recruitment patterns between these seemingly distinct behaviors (*Buford and Smith, 1990*; *Neptune et al., 2000*). We are interested in understanding how the *Drosophila* larva executes multiple behaviors, in particular forward versus backward crawling (*Carreira-Rosario et al., 2018*). Are there different MNs used in each behavior? Are the same MNs used but with distinct patterns of activity determined by premotor input? A rigorous answer to these questions requires both comprehensive anatomical information – that is a PMN/MN connectome – and the ability to measure rhythmic neuronal activity and perform functional experiments. All of these tools are currently available in *Drosophila*, and here we use them to characterize the neuronal circuitry used to generate forward and backward locomotion.

The *Drosophila* larva is composed of 3 thoracic (T1-T3) and nine abdominal segments (A1-A9; *Figure 1A*), with sensory neurons extending from the periphery into the CNS, and motor neurons extending out of the CNS to innervate body wall muscles. Most segments contain 30 bilateral body wall muscles that form 'spatial muscle groups' based on common location and orientation: dorsal longitudinal (DL; includes previously described DA and some DO muscles), dorsal oblique (DO), ventral longitudinal (VL), ventral oblique (VO), ventral acute (VA) and lateral transverse (LT) (*Figure 1B*) (*Crossley, 1978*; *Hooper, 1986*; *Bate, 1990*). Using these muscles, the larval nervous system can generate both forward and backward locomotion (reviewed in *Kohsaka et al., 2017*; *Clark et al., 2018*). Forward crawling behavior in larvae involves a peristaltic contraction wave from posterior to anterior segments; backward crawling entails a posterior propagation of the contraction wave (*Crisp et al., 2008*; *Dixit et al., 2008*; *Berni et al., 2012*; *Gjorgjieva et al., 2013*; *Heckscher et al., 2015*; *Pulver et al., 2015*; *Loveless et al., 2018*; *Kohsaka et al., 2019*) (*Figure 1A*).

There are ~30 bilateral pair of MNs in each segment: 26 pair of type Ib MNs with big boutons that typically innervate one muscle; two pair of type Is MNs with small boutons that innervate large groups of dorsal or ventral muscles; one or two type III insulinergic MNs innervating muscle 12; and three type II ventral unpaired median (VUM) MNs that provide octopaminergic innervation to most muscles (*Table 1*) (*Gorczyca et al., 1993*; *Landgraf et al., 1997*; *Hoang and Chiba, 2001*; *Landgraf et al., 2003*; *Choi et al., 2004*; *Mauss et al., 2009*; *Koon et al., 2011*; *Koon and Budnik, 2012*; *Arzan Zarin and Labrador, 2019*). Elegant pioneering work showed that type Ib MNs innervating muscles in the same spatial muscle group typically project dendrites to the same region of the dorsal neuropil, creating a myotopic map (*Landgraf et al., 1997*; *Mauss et al., 2009*). Several MNs have been shown to be rhythmically active during larval locomotion (*Heckscher et al., 2012*; *Zwart et al., 2016*), but only a few of their premotor inputs have been described (*Kohsaka et al., 2014*; *Heckscher et al., 2015*; *Fushiki et al., 2016*; *Hasegawa et al., 2016*; *Zwart et al., 2016*; *Takagi et al., 2017*; *Carreira-Rosario et al., 2018*; *Kohsaka et al., 2019*). Some excitatory PMNs are involved in initiating activity in their target MNs (*Fushiki et al., 2016*; *Hasegawa et al., 2016*; *Zwart et al., 2016*; *Takagi et al., 2017*; *Carreira-Rosario et al., 2018*), while some inhibitory PMNs limit the duration of MN activity (*Kohsaka et al., 2014*; *MacNamee et al., 2016*; *Schneider-Mizell et al., 2016*) or produce intrasegmental activity offsets (*Zwart et al., 2016*). Interestingly, some PMNs are active specifically during forward locomotion or backward locomotion (*Kohsaka et al., 2014*; *Heckscher et al., 2015*; *Fushiki et al., 2016*; *Hasegawa et al., 2016*; *Takagi et al., 2017*; *Carreira-Rosario et al., 2018*; *Kohsaka et al., 2019*). Yet a comprehensive map of the activity and connectivity of the PMN-MN-muscle network, which is essential for a full understanding of how locomotor behavior is generated, remains unknown.

Here, we address the question of how the same MNs and muscles generate two distinct behaviors: forward and backward locomotion. There are multiple mechanisms that could generate different forward and backward locomotor behaviors. (1) Forward and backward locomotion may use the same intrasegmental contraction patterns, and only the direction of the wave changes. (2) Different muscles/MNs could be used in each behavior. (3) One or more muscles/MNs may show a different

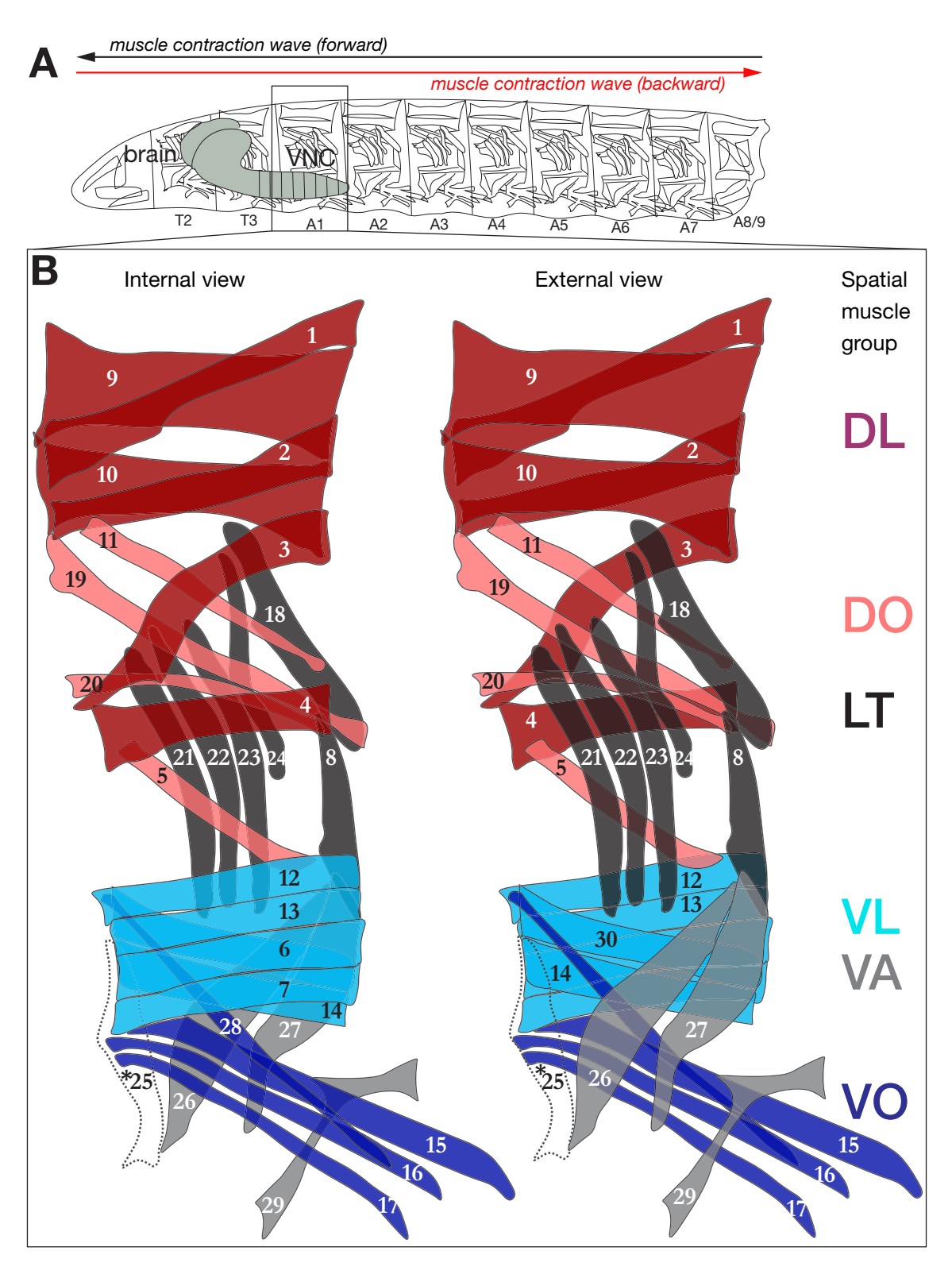

**Figure 1.** Schematic depiction of the larval neuromuscular system. (**A**) *Drosophila* larva contain three thoracic and nine abdominal segments, the muscles of which are innervated by MNs located in the corresponding thoracic and abdominal segments of the CNS. (**B**) Schematic of the 29 muscles of abdominal segments (A1) from internal and external view. Segments A2-A6 are similar to A1, with the exception that they have a muscle 25 (dashed line, asterisk) here overlaid on the A1 muscle pattern.

**Table 1.** Motor neurons present in the CATMAID reconstruction.

All MNs were identified in the first abdominal segment on both left and right sides, with the exception of MN25 which is not present in A1 and thus annotated in A2. See text for abbreviations.

| Spatial muscle group | Nerve | Motor neurons (synonyms) | Target muscles (synonyms) | Synapse Type |
|---|---|---|---|---|
| DL | ISN$^{DM}$ | MN1 (aCC) | 1 (DA1) | Ib |
| DL | ISN$^{DM}$ | MN2 (U3) | 2 (DA2) | Ib |
| DL | ISN$^{DM}$ | MN3 (U4) | 3 (DA3) | Ib |
| DL | ISN$^{DM}$ | MN4 (U5) | 4 (LL1) | Ib |
| DL | ISN$^{DM}$ | MN9 (U1) | 9 (DO1) | Ib |
| DL | ISN$^{DM}$ | MN10 (U2) | 10 (DO2) | Ib |
| DO | ISN$^{L}$ | MN11 | 11 (DO3) | Ib |
| DO | ISN$^{L}$ | MN19 | 19 (DO4) | Ib |
| DO | ISN$^{L}$ | MN20 | 20 (DO5) | Ib |
| DO | SNa | MN5 (LO1) | 5 (LO1) | Ib |
| VL | ISNb | MN6/7 (RP3) | 6/7 (VL3/VL4) | Ib |
| VL | ISNb | MN12 (V-MN) | 12 (VL1) | III |
| VL | ISNb | MN13 (MN-VL2) | 13 (VL2) | Ib |
| VL | ISNb | MN14 (RP1) | 14 (VO2) | Ib |
| VL | ISNb | MN30 (RP4) | 30 (VO1) | Ib |
| VA | SNc | MN26 | 26 (VA1) | Ib |
| VA | SNc | MN27 | 27 (VA2) | Ib |
| VA | SNc | MN29 | 29 (VA3) | Ib |
| VO | ISNd | MN15/16 (MN-VO4/5) | 15/16 (VO4/VO5) | Ib |
| VO | ISNd | MN15/16/17 (MN-VO4-6) | 15/16/17 (VO4/VO5/VO6) | Ib |
| VO | ISNb | MN28 | 28 (VO3) | Ib |
| T | SNa | MN8 (SBM) | 8 (SBM) | Ib |
| T | SNa | MN21/22 (LT1/LT2) | 21/22 (LT1/LT2) | Ib |
| T | SNa | MN22/23 (LT2/LT3) | 22/23 (LT2/LT3) | Ib |
| T | SNa | MN23/24 (LT3/LT4) | 23/24 (LT3/LT4) | Ib |
| T | ISN$^{L}$ | MN18 | 18 (DT1) | Ib |
| T | TN | MN25 (VT1) | 25 (VT1) | Ib |
| Broad | ISN$^{DM}$ | MNISN (RP2) | 1/2/3/4/9/10/11/[18]/19/20 (DA/DO) | Is |
| Broad | ISNb | MNISNb/d (RP5) | 6/7/12/13/14/15/16/30 (VL/VO) | Is |
| Broad | SNa | MNSNa-II (VUM) | 21/22/[23/24/25] (LT) | II |
| Broad | ISN$^{DM}$ | MNISN-II (VUM) | 1/2/3/4/9/10/11/18/19/20 (DA/DO) | II |
| Broad | ISNb | MNISNb/d-II (VUM) | 12/13/14/15/16/17/30 (VL/VO) | II |

time of recruitment in each behavior. (4) PMNs or MNs could have asymmetric morphology along the anteroposterior body axis, resulting in a different time of recruitment in each behavior. (5) One or more PMNs could be active only in forward or backward locomotion, changing the phase relationship of their target MNs. Here we use pan-muscle activity imaging, comprehensive TEM reconstruction of all MNs and well-connected PMNs, functional optogenetics, and development of a recurrent network model to sequentially test each of these hypotheses.

## Results

### All body wall muscles are activated during forward and backward locomotion

Intrasegmental differences between forward and backward locomotion could be due to recruitment of different muscles for each behavior, or changes in the timing of muscle recruitment. To distinguish between these mechanisms, we performed ratiometric calcium imaging to map the activation onset of each body wall muscle during forward and backward locomotion. To date only muscle contraction data have been reported, not muscle activity data, and only for five of the 30 body wall muscles (5, 9, 12, 21, and 22), showing that individual longitudinal muscles contract prior to individual transverse muscles during forward locomotion (*Heckscher et al., 2012*; *Zwart et al., 2016*). Muscle contraction could occur passively due to biomechanical linkage between adjacent muscles, so it may not be a perfect substitute for directly measuring muscle activity. Conversely, elevated GCaMP fluorescence may be insufficient to trigger muscle contraction, but it is a better proxy for monitoring excitatory inputs than is muscle contraction.

Here, we used GCaMP6f/mCherry live imaging to measure the activation time of individual body wall muscles in the abdominal segments during forward and backward locomotion. We expressed GCaMP6f and mCherry using the muscle line *R44H10-LexA*, which has variable expression in sparse to dense patterns of muscles. For this experiment we analyzed larvae with dense muscle expression. We imaged both forward and backward locomotion in 1 st and 2nd instar larvae (a representative animal shown in *Figure 2A,D*). We found that an increased GCaMP6f signal correlated with muscle contraction during both forward and backward locomotion (representative examples of muscle 6 shown in *Figure 2B,E*). Most importantly, all imaged muscles showed a significant rise in GCaMP6f fluorescence during forward and backward locomotion (*Figure 2C,F*; *Videos 1* and *2*). In addition, because each type Ib MN typically innervates a single muscle, we can use muscle depolarization as a proxy for the activity of its innervating MN. We conclude that all MNs and their target muscles are activated during forward and backward locomotion.

### A small number of muscles are differentially recruited during forward and backward locomotion

All muscles are recruited in both forward and backward locomotion, leading to the hypothesis that any possible difference in forward and backward locomotion should result from different muscle recruitment times. If so, we predicted VO and DO muscles to behave differently in forward versus backward, because they have asymmetric localization along the anteroposterior axis (*Figure 1*). To test this hypothesis, we embedded the multidimensional crawl cycle data in two-dimensional space using principal component analysis (PCA)(*Lemon et al., 2015*). We aligned crawl trials by finding peaks in this 2D space which corresponded to the highest contraction amplitude of the most muscles in a given crawl (*Figure 3—figure supplement 1*). Muscle activity appeared as a continuum with the sequential recruitment of individual muscles, yet hierarchical clustering of the mean activity of each muscle during forward and backward crawling revealed four groups of co-active muscles for both behaviors (*Figure 3A–E*; *Table 2*). We call these co-activated muscle groups F1-F4 for forward and B1-B4 for backward crawling. Overall, we analyzed 27 muscles during forward locomotion and 25 muscles during backward locomotion (the missing muscles were too tightly packed to extract clear activity profiles). The activation time of each co-activated muscle group was more coherent than the time of their inactivation (*Figure 3B–E*). Notably, these co-activated muscle groups do not fully match previously identified spatial muscle groups (compare *Figures 1B* and *3F,G*).

We found that the largest change in recruitment time between forward and backward locomotion was in six muscles: the three muscles in the VO spatial muscle group (muscles 15–17), and muscles 2,

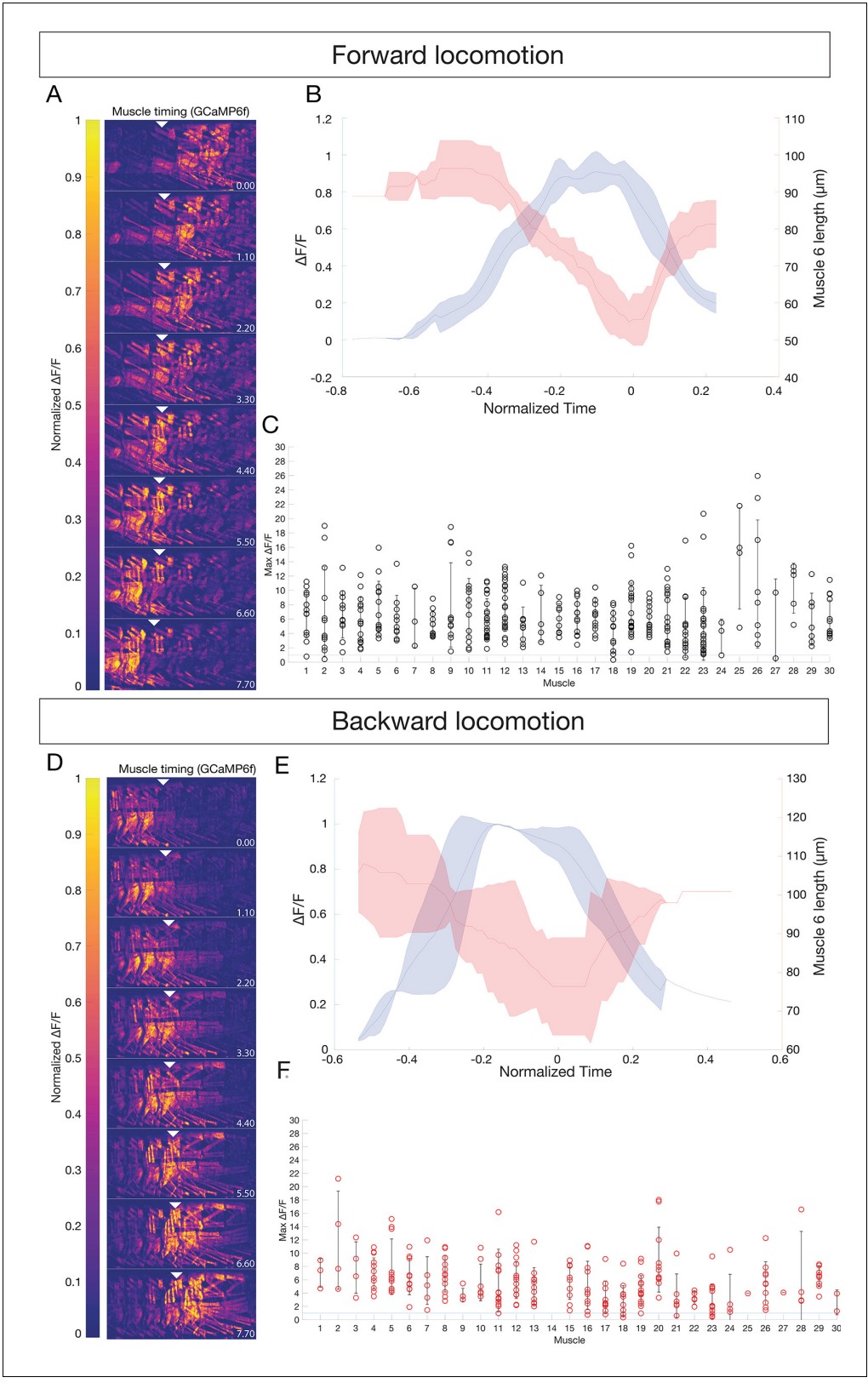

**Figure 2.** All body wall muscles are utilized during forward and backward locomotion. (**A,D**) Sequential images of muscle GCaMP6f ΔF/F signal during forward (**A**) or backward (**D**) locomotion. GCaMP6f levels were normalized to mCherry. Anterior to left, dorsal up; time in seconds. Genotype: *GMR44H10-LexA lexAOP-GCaMP6f; -LexA lexAOP–mCherry*. Arrowheads mark the same segment at each timepoint; A2 in (**A**) and A4 in (**D**). (**B,E**) Mean
*Figure 2 continued on next page*

*Figure 2 continued*

calcium transient (blue) vs mean muscle length (red) measurements for muscle six during forward (**B**) or backward (**E**) locomotion. N = 3 segments. $T_0$ was set as the point of maximum contraction as determined by muscle length for each crawl. Shaded bars represent standard deviation. (**C,F**) All observed muscles show calcium transients greater than 100% ΔF/F during forward (**C**) or backward (**F**) locomotion. Each dot represents the maximum GCaMP ΔF/F signal in the indicated muscle during a single crawl, normalized to mCherry. Error bars represent standard deviation. Muscle names as in *Figure 1*.

11, and 18 (each in a different spatial muscle group) (*Figure 3H,I*; *Figure 3—figure supplement 2*). The VO spatial muscle group (muscles 15–17) switched from late activity during forward locomotion (F3) to early activity during backward locomotion (B1), whereas the three other neurons switched from early activity during forward locomotion (F1/2) to late activity during backward locomotion (B3/4) (*Figure 3H,I*). Other spatial muscle groups typically did not change their timing of activation; for example longitudinal muscles tended to be active early and transverse muscles activated late in both behaviors (*Figure 3F,G*), consistent with prior reports tracking single muscles within each group (*Heckscher et al., 2012*; *Zwart et al., 2016*). We conclude that differences between forward and backward locomotor behaviors may arise from the relatively small number of MN/muscles that show differential recruitment during each behavior. Our results raise two new questions. (1) What mechanisms produce co-active muscle groups? (2) What mechanism produce the differential timing of the VO and 2/11/18 muscles in forward and backward locomotion? Answering these questions will help determine how the same MNs and muscles can generate two different locomotor behaviors.

## TEM reconstruction of motor neurons in A1 segment

To understand how motor patterns are generated, it is essential to map connectivity from muscles to MNs, and from MNs to PMNs. In this section, we fully reconstruct all 31 MNs in segment A1, and below we fully reconstruct 118 PMNs providing input to these MNs. These data on neuronal morphology, synapse localization, and connectivity will generate testable hypotheses for how different motor behaviors are generated.

To date, only less than half of the 31 abdominal MNs have been fully reconstructed at synapse level resolution (1, 5, 6/7, 9,10,18, 21/22, 22/23, 23/24, 30, MNISN, and MNISNb/d) (*Heckscher et al., 2015*; *Fushiki et al., 2016*; *Schneider-Mizell et al., 2016*; *Zwart et al., 2016*; *Takagi et al., 2017*; *Carreira-Rosario et al., 2018*; *Kohsaka et al., 2019*). Here, we identify, comprehensively reconstruct, and map dendritic postsynaptic sites for all remaining A1 MNs, which can be used as a proxy for other abdominal segments. We reconstructed 16 pair of type Ib MNs, including MNs innervating muscles that are differentially active in forward versus backward locomotion (*Figure 4*, red outlines). We identified one pair of type III MNs that target muscle 12, and the three unpaired midline octopaminergic MNs (VUMs) (*Figure 4*; *Table 1*). In subsequent analyses, we did not include the neuromodulatory VUM MNs due to their relatively undifferentiated state (few postsynapses). In addition to the two previously identified Is MNs (MNISN and MNISNb/d), the presence of yet another type Is MN has been suggested (*Hoang and Chiba, 2001*), but we did not find it in the TEM volume; it may be late-differentiating or absent in A1. We linked all bilateral MNs in the TEM volume to their muscle target by matching the dendritic morphology in the EM reconstruction to the dendritic morphology determined experimentally (*Landgraf et al., 1997*; *Landgraf et al., 2003*; *Mauss et al., 2009*) (*Figure 4*; *Figure 4—figure supplement 1*; *Table 1*). A dataset of all

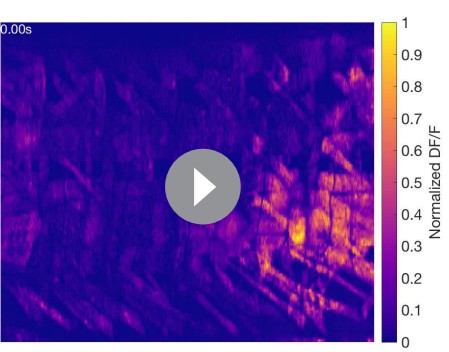

**Video 1.** GCaM6f muscle activation during forward locomotion in a *Drosophila* second instar larva. Dorsolateral view; anterior left.
https://elifesciences.org/articles/51781#video1

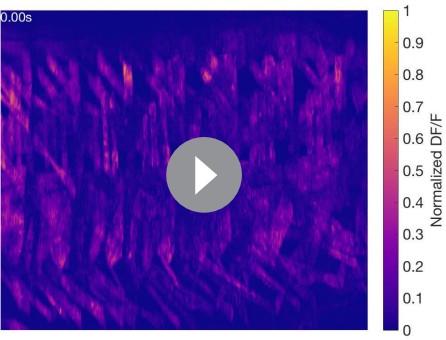

**Video 2.** GCaM6f muscle activation during backward locomotion in a Drosophila second instar larva. Dorsolateral view; anterior left.
https://elifesciences.org/articles/51781#video2

MNs that can be opened in CATMAID (*Saalfeld et al., 2009*) is provided as *Supplementary file 1*. Note that the transverse nerve MN (MN25-1b) is only present in the A2-A7 segments (*Hessinger et al., 2017*), so we traced it in A2. We found that all MNs had a dense array of postsynapses on their dendritic projections, but unlike *C. elegans* (*Wen et al., 2012*), we observed no presynaptic contacts to other MNs or interneurons (*Figure 4—figure supplement 1*). In conclusion, we have successfully identified and reconstructed, at single synapse-level resolution, all differentiated MNs in segment A1 of the newly hatched larval CNS. This is a pre-requisite for mapping the location of postsynaptic sites, as well as for mapping PMN-MN connectivity (below).

## Co-active motor neurons have dispersed postsynaptic sites within the dorsal neuropil

Motor neurons innervating a single spatial muscle group target their dendrites to a similar region of the neuropil, creating a myotopic map in the neuropil (*Landgraf et al., 2003*; *Mauss et al., 2009*). Here we validate this conclusion at the level of postsynapse neuropil localization, and determine whether similar clustering is found for MNs in a co-active muscle group. To begin, we calculated pairwise synapse similarity scores (*Schlegel et al., 2016*) for MNs in the left and right A1 hemisegments and observed highly similar postsynapse clustering within the neuropil volume (Pearson correlation coefficient, $r = 0.97$), which we averaged for subsequent analysis. This validated the quality and reproducibility of the MN dendritic synapse data and highlighted the stereotypy of MN postsynaptic locations in the neuropil. Next, we performed unbiased clustering of MNs based on postsynaptic synapse similarity, and found a highly ordered hierarchical relationship between postsynapse localization and innervation of spatial muscle groups (*Figure 5A*). We also found that MNs innervating each spatial muscle group have different postsynaptic distributions along two axes (two sample Kolmogorov-Smirnov test; p<0.05) (*Figure 5B*). Our data strongly support and extend prior work showing that MNs innervating spatial muscle groups form a myotopic map in the neuropil, providing a first layer of functional organization of the motor neuropil.

Next we asked: do co-active MNs generate a 'co-active' neuropil map? Interestingly, MNs innervating each forward or backward co-active muscle group had distinct postsynapse density maxima along the mediolateral axis, and often along a second axis (either dorsoventral or anterioposterior) (*Figure 5C,D* arrowheads). Although the maxima are different along each axis, there is considerable overlap, such that there are only a few regions of unique postsynapse targeting (*Figure 5C,D* asterisks). We conclude that there is an ordered distribution along the mediolateral axis of postsynapses from MNs that innervate distinct forward or backward co-active muscles. Whether these distinct maxima or unique neuropil locations of MN postsynapses are functionally important for generating locomotor behavior remains a question for future functional studies.

We next addressed the question of how specific muscles can be recruited at different times during forward and backward locomotion, as seen for muscles 2, 11, 18, and the three VO muscles. We first ask whether each of the MNs innervating these six muscles target their postsynapses to a different region of the neuropil compared to the surrounding neurons in the same co-active muscle group. There are single MNs innervating each of the muscles 2, 11, and 18 (MN2, MN11, MN18); and two MNs innervating the three VO muscles (MN15/16, MN15/16/17). We found that MNs innervating muscles 11, 18 and the VOs each targeted postsynapses to a unique region of the neuropil. The VO MNs have a medial postsynaptic distribution not seen in other neurons in the same co-active muscle group (*Figure 5E*); MN11 has a synapse localization maxima that is distinct from other co-active neurons (*Figure 5F*); and MN18 has a posterior postsynaptic distribution not seen in other co-active neurons (*Figure 5G*). In contrast, MN2 did not have a distinct distribution in any axis

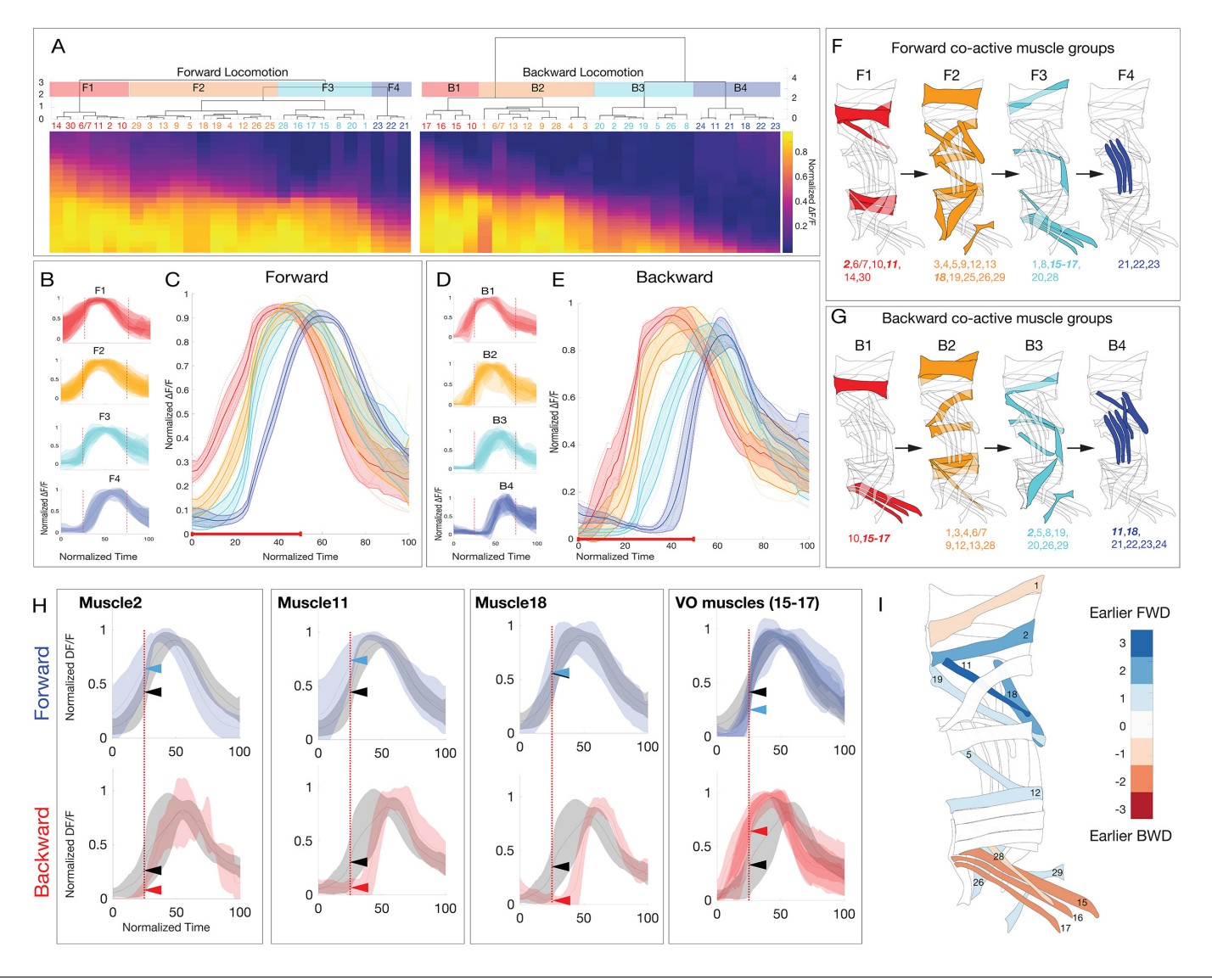

**Figure 3.** Larval body wall muscles form four co-activated muscle groups during forward and backward locomotion. (A) Hierarchical clustering of mean activity for all observed muscles yields four co-activated muscle groups during forward locomotion (F1–F4) and a different group of four during backward locomotion (B1–B4). Heatmaps represent the mean range-normalized calcium activity of each muscle (n > 3 crawl bouts for each muscle, with a total of 337 individual muscles analyzed across 23 crawls for forward and 188 individual muscles analyzed across 14 crawls for backward locomotion). Muscles 6/7 are grouped because they are both innervated by the same MN. Clustering was performed only on the first half of the crawl cycle to determine the onset time for each co-activated muscle group. Cluster number was determined by visual inspection of the dendrogram as well as the gap-criterion optimal cluster number. (B,D) Plots of average muscle activity for muscles in each forward or backward co-activated muscle group. Error bars represent the standard deviation of individual muscles. (C,E) Plots of average forward or backward co-activated muscle group activity timing. Error bars represent the standard deviation of the average muscle activity of each muscle in a given co-activated muscle group. Dotted lines represent the average muscle activity for each muscle in a given co-activated muscle group. Red line along the x-axis represents the fraction of the crawl cycle that was used for clustering. (F,G) Schematic representation of the co-activated muscle group for forward or backward locomotion. (H) Plots of muscles that are differentially active during forward or backward crawling. For forward panels, the gray trace represents the mean calcium activity of all muscles during a forward crawl, while the blue trace represents the activity of the indicated muscle. For backward panels, the gray trace represents the mean calcium activity of all muscles during a backwards crawl, while the red trace represents the mean calcium activity of the indicated muscle. Error bars represent standard deviation. Dotted red line marks t = 25 (normalized time). Arrows represent the normalized ΔF/F of the two traces at t = 25. (I) Heatmap illustrating differential activity of muscles during forward versus backward crawling. Blue, or positive values indicate a given muscle is active earlier during forward crawling, while red or negative values indicate a given muscle was active earlier during backward crawling.

The online version of this article includes the following figure supplement(s) for figure 3:

**Figure supplement 1.** PCA-based alignment of crawl cycles.

*Figure 3 continued on next page*

*Figure 3 continued*

**Figure supplement 2.** Muscles recruited at similar and different phases of the forward and backward crawl cycle.

(*Figure 5H*), showing that differential recruitment can occur despite the intermingling of postsynapses with other MNs in the same co-active group. We conclude that differential MN postsynaptic localization is not required for generating differential muscle recruitment; and that MN postsynapse localization alone is insufficient to explain differential muscle recruitment. A full understanding requires characterization of PMN-MN connectivity.

## TEM reconstruction of 118 premotor neurons reveals premotor neuron pools targeting each group of co-active motor neurons

There are two hypotheses for how co-active MNs are recruited. Each pool of co-active MNs may be innervated by a distinct pool of PMNs (labeled line), or alternatively each pool of co-active MNs may be innervated by different combinations of PMNs (combinatorial code). To distinguish between these models, we identified and reconstructed all PMNs in the TEM volume with dense monosynaptic contacts to A1 MNs. The names of each premotor neuron along with previously published synonyms is given in *Supplementary file 2*. This included local premotor neurons with somata in A1 as well as neurons from adjacent segments with dense connectivity to A1 MNs. PMNs were identified by contributing greater than 1% (and >4 synapses) of the total input onto a given MN (*Figure 6—figure supplement 1*; see Materials and methods for additional PMN selection criteria). We identified 118 bilateral PMNs (236 total) with connectivity to A1 MNs (*Table 3*; see Materials and methods for selection criteria).

As this is the first comprehensive characterization of larval PMNs, we first quantified key features of this population. The morphology of each of the 118 pair of PMNs is shown in *Figure 6—figure supplement 2*. We found that PMN presynapses were enriched in the dorsal neuropil, as expected, and PMN postsynapses were distributed throughout the neuropil (*Figure 6A,B*). Each PMN synapsed with an average of 8.0 MNs (*Figure 6C*), while each MN received input from an average of 32.5 PMNs (*Figure 6D*). PMNs made 7495 synapses on A1 MNs, which accounted for 12.7% of PMN output and 76% of the total A1 MN input (excluding A2 MN-25) (*Figure 6E,F*). The fraction of total PMN to A1 MNs was highly variable, with some PMNs having as little of 0.6% of their output onto A1 MNs while others had as much as 99.6% (*Figure 6E*). Conversely, MNs received 59.6% to 97.9% of their total inputs from these 118 PMN pairs (excluding RP3 which has most of its postsynapses anterior to A1) (*Figure 6F*). In addition, most PMNs projected contralaterally, had local arbors, and had postsynaptic inputs on their more proximal processes (*Figure 6G–I*). Neurotransmitter

**Table 2.** Co-activated muscle groups during forward or backward locomotion.
There are four co-activated muscle groups during backward and forward locomotion, but the muscles in each group differ in forward versus backward locomotion. Note that backward locomotion is not simple a reverse of the pattern seen in forward locomotion. This represents the most common activation sequences, although there is some variation, particularly during the fastest locomotor velocities.

| Forward | Co-activated muscles |
| --- | --- |
| F1 | 2,6,10,11,14,30 |
| F2 | 3,4,5,9,12,13,18,19,25,26,29 |
| F3 | 1,8,15,16,17,20,28 |
| F4 | 21,22,23 |
| **Backward** | **Co-activated muscles** |
| B1 | 10,15,16,17 |
| B2 | 1,3,4,6,9,12,13,28 |
| B3 | 2,5,8,19,20,26,29 |
| B4 | 11,18,21,22,23,24 |

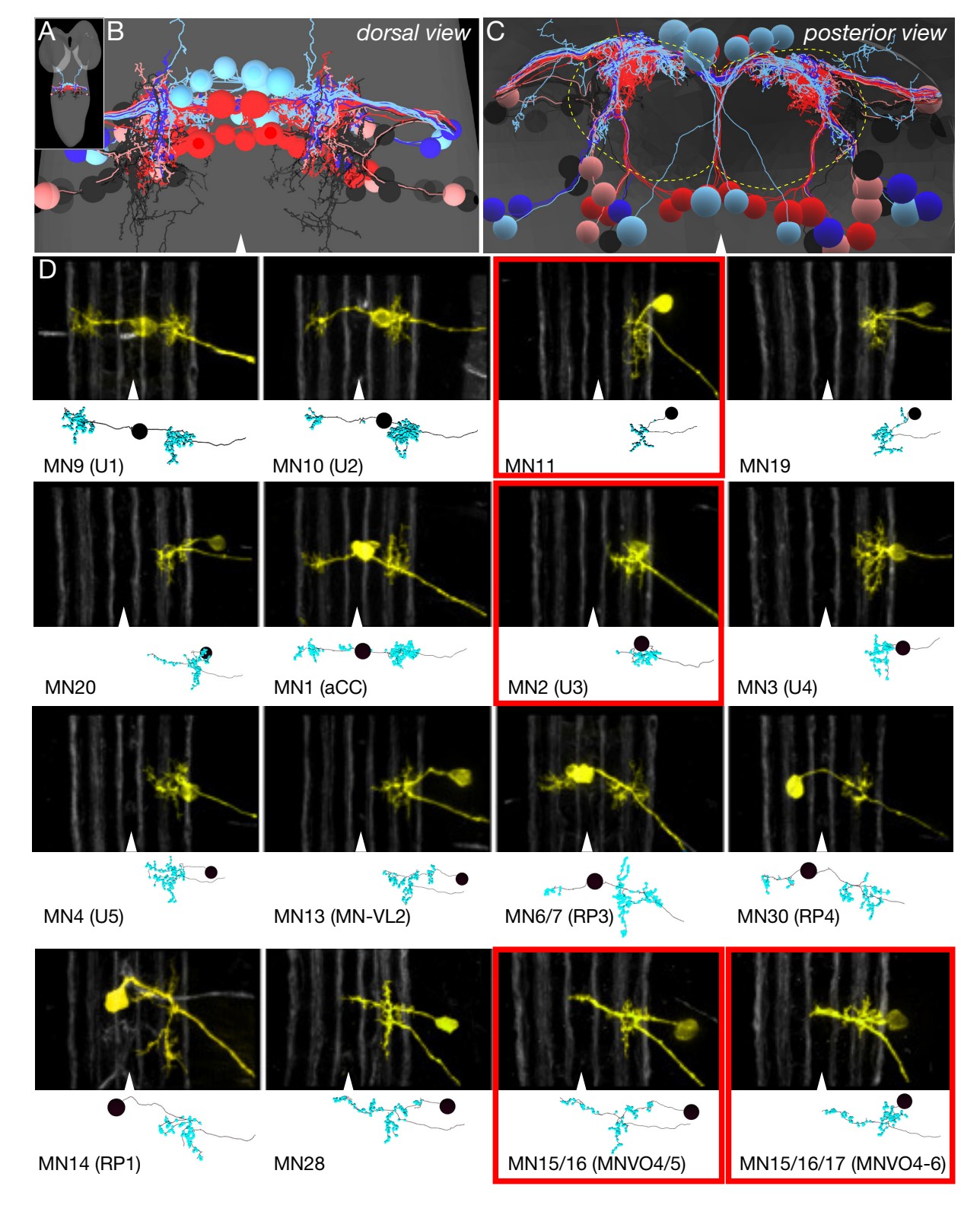

**Figure 4.** Identification of all motor neurons in segment A1 in the TEM volume. (**A**) Dorsal view of the TEM reconstruction of the L1 CNS (gray shading) showing all bilateral MNs in A1 reconstructed at single synapse level. The one intersegmental dendrite is from RP3 in A1; it is not observed in other abdominal segments. (**B**) Dorsal view of centered on the A1 segment; midline, arrowhead. MNs are color-coded as in *Figure 1A*: DL MNs (red), DO MNs (light red), VL MNs (light blue), VO MNs (dark blue), LT MNs (black), VA MNs (gray). (**C**) Posterior (cross-section) view of the neuropil (outlined) and

*Figure 4 continued*

cortex in A1. Note the MN dendrites target the dorsal neuropil. Dorsal, up; midline, arrowhead; neuropil border, dashed outline. (**D**) Representative images showing the morphological similarity between MNs identified in vivo by backfills (*Mauss et al., 2009*) versus the most similar MN reconstruction from the TEM volume. The top section in each panel shows the morphology of the MN dendrites based on in vivo backfills; used with permission); six distinct Fas2 fascicles (three per hemisegment) are shown in white; midline, arrowhead. The bottom section shows MN dendrite morphology reconstructed from the TEM volume in A1. MNs highlighted in red boxes show differential recruitment timing during forward versus backward locomotion.

The online version of this article includes the following figure supplement(s) for figure 4:

**Figure supplement 1.** Reconstruction and identification of A1 MNs in the TEM volume.

expression is known only for a subset of PMNs (*Supplementary file 3*), so we screened for Gal4 lines with sparse expression patterns, performed MultiColorFlpOut (*Nern et al., 2015*) to match their morphology to individual PMNs, and mapped neurotransmitter expression (*Figure 6J*, *Supplementary file 3*). A file that can be opened in CATMAID showing all 118 bilateral PMNs is provided as *Supplementary file 4*. In conclusion, we have identified a large majority of the PMN inputs to the MN population in segment A1, and mapped neurotransmitter expression for the majority of these PMNs.

Following our characterization of the PMN population, we next asked whether there are PMNs dedicated to innervating individual spatial or co-active MNs, or MNs differentially recruited during forward and backward locomotion. We identified PMNs innervating MNs of a single spatial muscle group (*Figure 7A*), as well as PMNs specifically innervating MNs in a single forward or backward co-active muscle group (*Figure 7B,C*, *Figure 7—figure supplement 1*). We found that 30 of the 118 PMNs innervated MNs in a single spatial muscle group (*Figure 7D*). Interestingly, a similar numbers of PMNs innervated MNs in a single co-active muscle group (*Figure 7E,F*). Thus, we have identified groups of PMNs that specifically innervate co-active MNs, consistent with a 'labeled line' model for generating motor output, yet we note that the majority of PMNs innervate MNs in multiple spatial or co-active muscle groups. Our data are consistent with both labeled line and combinatorial codes for driving co-active motor neuron output; functional studies will be necessary to determine their relative importance.

Next we examined the five MNs that showed differential recruitment during forward versus backward locomotion (MNs 2, 11, 18, and two VOs), to see if they were selectively innervated by 'labeled lines' of PMNs. We found that all PMNs innervate multiple MNs, and there is no evidence for specific PMNs innervating specific MNs, whether they are differentially recruited or not (*Figure 7G*). We conclude that PMN combinatorial coding is likely to generate the observed MN differential recruitment during forward and backward locomotion.

Lastly, we tested whether our connectomic data could be used to predict the timing of MN recruitment. We found that the PMN A27h (*Figure 7H*) is strongly connected to MNs in co-active group F3 (*Figure 7H'*), so we asked whether A27h was recruited after the U1-U5 MNs in co-active groups F1/F2. Indeed, dual color calcium imaging showed that the F1/F2 MNs were active prior to the F3 PMN A27h (*Figure 7H''*). These results support the use of connectivity to predict MN recruitment times.

## Neuronal asymmetry may generate different muscle recruitment times during forward and backward locomotion

Asymmetric dendrite morphology can be an important determinant of neuronal function, such as in the direction-selective T4/T5 neurons in the adult visual system (*Fisher et al., 2015*). Similarly, dendritic asymmetry along the anteroposterior axis may help generate temporally distinct recruitment that we observe during forward and backward locomotion. We examined the morphology of differentially recruited MNs and found that MN18, but not the others, is highly asymmetric (*Figure 8A–E*). The asymmetric distribution of postsynaptic sites on MN18 should lead to its earlier activation during forward than backward locomotion. This is consistent with its activity pattern deciphered using muscle calcium imaging during these behaviors (*Figure 3*). We also observed anterior/posterior asymmetry in multiple PMNs. For example, A02i and A03a4 have axons extending 1–2 segments anterior of the dendrites; A03a5 has axons projecting 1–2 segments posterior of the dendrites; whereas A03g

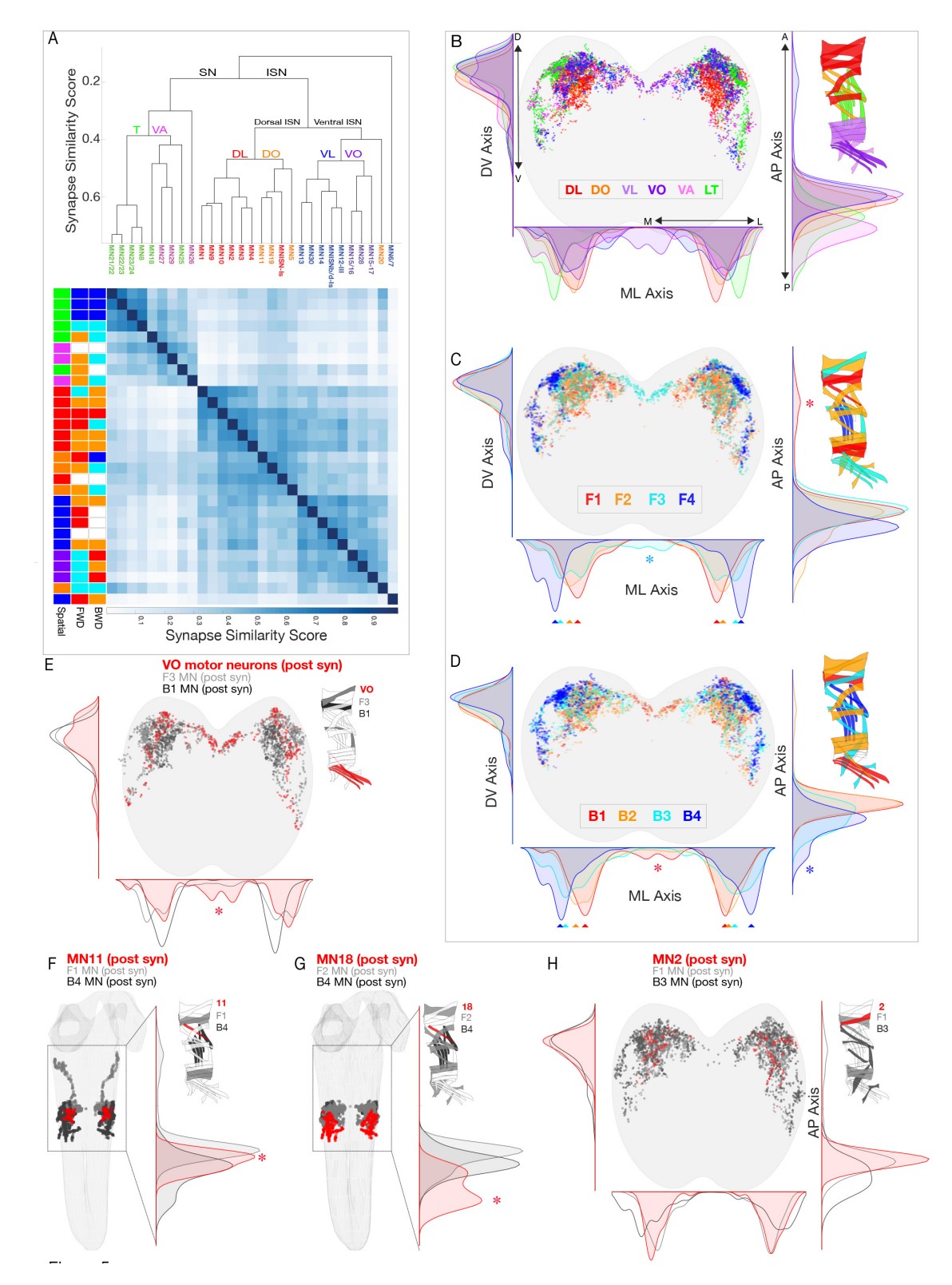

**Figure 5.** Motor neurons innervating spatial muscle groups or co-activated muscle groups have post-synapses in distinct regions of neuropil. (A) Hierarchical clustering of MNs by their synapse similarity score reveals MN myotopic organization. To generate a similarity matrix, pairwise synapse similarity scores were generated separately for MNs exiting the left A1 nerve and right A1 nerve. The pairwise similarities for the left and right pools of MNs were highly correlated (*r* = .95); clustering was performed on the average of the left and right similarity matrices. (B–D) Spatial distributions of

*Figure 5 continued on next page*

*Figure 5 continued*

postsynaptic sites for the indicated spatial or co-active MNs. Plots are 1D kernel density estimates for the mediolateral (ML), dorsoventral (DV) and anteroposterior (AP) axes. Asterisks, postsynapses from a single group that are enriched in a specific region of the neuropil. (E–H) Spatial distribution of postsynapses for the differentially recruited MNs (red) compared to the MNs in their forward or backward co-active group (gray or black, respectively).

is a symmetric PMN (*Figure 8F–H*). Due to the opposite direction of wave propagation in backward and forward locomotion, the asymmetric PMNs are likely to contribute to the differential MN/muscle recruitment in forward and backward locomotion.

## A recurrent network model that generates the observed forward and backward pattern of muscle activity

Recurrent interactions among PMNs have been shown to control the timing of the muscle outputs of central pattern generator circuits in a variety of organisms (*Marder and Bucher, 2001*; *Grillner, 2003*). We hypothesized that these types of interactions are responsible for the timing of muscle activation during *Drosophila* larval forward and backward crawling. To assess whether the reconstructed PMN connectome is capable of producing the observed timing of MN/muscle activation, we developed a recurrent network model of two adjacent segments. Previous models have focused on wave propagation during forward and backward crawling by modeling the average activity of excitatory and inhibitory subpopulations in each segment (*Gjorgjieva et al., 2013*; *Pehlevan et al., 2016*). Access to the detailed connectivity of PMNs and MNs (*Supplementary file 5* and *Supplementary file 6*), as well as knowledge of the activation patterns of different co-activated muscle groups, allowed us to develop a substantially more detailed model whose circuitry was constrained to match the TEM reconstruction. For PMNs whose neurotransmitter identity we could determine, we also constrained the signs (excitatory or inhibitory) of connection strengths in the model. The firing rates of PMNs and MNs were modeled as simple threshold-linear functions of their synaptic inputs, and model parameters were adjusted to produce target MN patterns of activity that matched the sequences identified during forward and backward crawling. These patterns were assumed to be evoked by external command signals, representing descending input to the PMNs, that differed for forward and backward crawling but did not themselves contain information about the timing of individual muscle groups. We also constrained the activity of two PMNs, A18b and A27h, that are known to be specifically active during backward and forward locomotion, respectively (*Fushiki et al., 2016*; *Carreira-Rosario et al., 2018*). We found that, although the connectivity among PMNs within a segment is sparse (roughly 7% of all possible pairwise connections), the observed connections are nonetheless sufficient to generate appropriately timed MN activity for the two distinct behaviors (*Figure 9A,B*; *Figure 9—figure supplement 1*). As has been described previously in other pattern-generating systems (*Prinz et al., 2004*), there is a space of models that is capable of producing the observed activity. We therefore analyzed the activity of neurons in an ensemble of models. In the models, distinct sequences of PMN activity for forward and backward locomotion tile the period of time over which MNs are active (*Figure 9C*; *Figure 9—figure supplement 1*). These sequences give rise to the distinct timing of MN activation during each behavior.

We used knowledge about the differential recruitment of two PMNs, A27h and A18b, to constrain our model. It is interesting to ask whether this constraint is required, or whether connectivity alone reveals this selectivity. When we constructed models lacking a penalty that enforces selective activation of these two neurons, we found that only A27h retained its selectivity (*Figure 9—figure supplement 2*). This suggests that the PMN-MN connectome is insufficient to capture the selectivity of A18b to backward locomotion, which is consistent with a recent study that showed that it could be activated by a descending neuron (not part of our analysis) specifically during backward locomotion (*Carreira-Rosario et al., 2018*). Characterizing and incorporating this descending circuitry will be important to refine future models.

Next we asked if the sequences of PMN activity predicted by the model are consistent with prior experimentally determined activity patterns. In our model, the PMN A14a is active at F1 and is inactive at F4 (*Figure 9C*). Similarly, experimental data show that A14a is inhibitory and is active during co-activated muscle group F1; and blocking A14a activity removes the contraction delay between muscles in co-activated muscle group F1 and F4 (*Zwart et al., 2016*), thereby validating our model. In our model, the PMNs A18b3 and A18a are both active during forward locomotion, but only A18a

**Table 3.** Premotor neurons innervating type Ib MNs Left column, spatial muscle groups named as in **Figure 1**. Middle column, type Ib MNs innervating 1–3 muscles in each muscle group (synonym, parentheses); the immature neuromodulatory VUMs are not shown. Right column, premotor interneurons innervating the indicated MNs. Premotor connectivity uncertain, parentheses.

| Muscle position | Motor Neurons | Pre-Motor Neurons |
|---|---|---|
| DL | MN1-Ib (aCC) | A27h, A18a, A18b, A03g, A31k, A31b, A06e, A23a, A02h, A10e, A03a1, A03a3, A05k, A07f2, DLN2, TJPMN Thoracic descending pre-longitudinals, T27Y, dsnPMN2, DLN1, A18neo. |
| DL | MN2-Ib (U3) | A01x2, A18a, A03a5, A31k, A31b, A23a, A02h, A03a3, A03a1, A10e, A10a , T27Y, dsnPMN2. |
| DL | MN3-Ib (U4) | A18a, A03a5 A03g, A31k, A31b, A06e, A02h, A02e, A02f, A03a3, A03a6, A03d/e, A03x-eghb, A07f2, A10a, A18neo. |
| DL | MN4-Ib (U5) | A03a5, A03g, A31k, A27l, A06l, A06m, A06g2, A02e, A02f, A03a6, A03a1, A03x-eghb, SePN02b, DLN2, Descending, pre RP3, A18neo. |
| DL | MN9-Ib (U1) | A01x2, A18a, A31k, A31b, A06x1, A27l, A23a, A02m, A02n, A02h, A03a1, A03a3, A03x-eghb, A03xyz, A05k, DLN2, DLN2, TJPMN, Tipsi, T27Y, dsnPMN2, DLN1, A18neo. |
| DL | MN10-Ib (U2) | A01x2, A18b, A08e1, A31k, A27j, A23a, A06a, A06x1, A02h, A02e, A02g, A10e, A03a1, A03a3, A03x, A03a4, A03d/e, A03x-eghb, VLELX4, Tipsi, dsnPMN2, DLN2, DLN1, A18neo, A18c. |
| DO | MN11-Ib | A31k, A06x1, A23a, A06a, A27l, T03g2, A03a1, A03a3, A03x-eghb. |
| DO | MN19-Ib | A27k, A18j, A18b, A18b3, T01d2, A31k, A27j, A23a, A06a, A06l, A06x1, A02f, A03a1, A03a3, T27Y, dsnPMN2, A27neo. |
| DO | MN20-Ib | A27h, A18j, A01c1, T01d2, T01d4, A19l, A06e, A03d/e, A27neo, a14neo, A03xyz, A26f. |
| DO | MN5-Ib (LO1) | A18b3, A18b2, A23a, A03a1, A03a3, A03a4, VLELX4, T27Y. |
| VL | MN6/7-Ib (RP3) | A18b3, A03a5, A27l, A06l, A06e, A02g, A02e, A03a4, T06WW, T06PP, Descending pre RP3. |
| VL | MN12-III (V-MN) | A27h, A03a5, A03g, A02g, A02e A27l, A06l,, A06e A03a6, A03a4, A03d/e, DLN1, Descending pre RP3. |
| VL | MN13-Ib (MN-VL2) | A27k, A03a5, A03g, A01d3, T01d4, A06l, A06a, A06e, A02g, A02e, A27l, A03a6, A03a4, A03x-eghb, A03d/e. |
| VL | MN14-Ib (RP1) | A27h, A18b2, A18b3, A27l, A06l, A02i, A03a4, A03a1, DLN1. |
| VL | MN30-Ib (RP4) | A18b3, A03a5, A01x2, A01d3,A01d4, A06e , A27l, A06l, A02g, A02e, A03a4, A03a6, A03x-eghb, A03d/e, A03SNC, A10a, A27Uniq, DLN1, A03xyz, SePN02b |
| VA | MN26-Ib | A27h, A01x3, A18f, A02j, A06e, A06l, A27l, T03g2, A03x-eghb, Descending neuron_SEZ, A03SNC, A03xKT, A03d/e, T11v, T27Y. |
| VA | MN27-Ib | A27h, A27k, A03g, A18j, A18f, A01x3, A01c1, A01c2, T01d2, T01d4, A06e, A06f, A19l, A14a, A31b, T03g2, A27n, A27neo, A03xKT, T11v, A26f. |
| VA | MN29-Ib | A01x3, A01x2, A01x3, T01d2, T01d4, A27l, A02g, A06e, T03g2, A27e2, A03a6, A03d/e, A10a, A27neo, T11v, A03SNC. |
| VO | MN15/16-Ib | A27h, A27k, A18b2, A06c, A06l, A06e, A02g, A02i, A03a6, DLN1. |
| VO | MN15/16/17-Ib | A27h, A03g, A06c, A06e, A27l, A02g, A02i, A01j, A27Uniq. |
| VO | MN28-Ib | A01x2, A27h, A18b2, A06c, A06l, A06e, A02g, A02i, A03a6. |
| T | MN8-1b (SBM) | A01c1, A01c2, A01d3, A27k, A03g, T01d1, A18j, A19l, A14a, A27n, A27e2, A27neo, A26f. |
| T | MN18-Ib | A01c1, A01c2, A01d3, A03g, A03o, A18j, A06a, A23a, A19l, A14a, A06x1, A02i, A01j, A27n, A10a, A10b, A27neo, T27Y, A26f. |
| T | MN21/22-Ib (LT1/LT2) | A01c1, A01c2, A27k, A03g, A18j, A18b2, T01d1, T01d2, A19l, A14a, A02i,A02f, A03xKT, T27Y, TGun, A27n, A27neo, A26f. |
| T | MN22/23-1b (LT2/LT3) | A01x, A01c1, A01c2, A27k, A03g, A09l, A18j, T01d1, T01d2, A01d3, A19l, A14a, A02f, A27n, A27neo, A27e2, T27Y, A26f. |
| T | MN23/24-1b (LT3/LT4) | A27k, A18j, A03g, A01c1, A01c2, T01d1, T01d2, A01d3, A19l, A27n, A27neo, A26f. |
| T | MN25-Ib (MN-VT1) | A01c1, A18a, A18b2, A18j, A18f, A27l, A14a, A19l, A02i, A31d, A03xKT, A05a. |
| DL/DO | MNISN (RP2) | A01x2, A18b, A03g, A31k, A27j, A27l, A02m, A02n, A02b, A06a, A23a, A03a1, A03a3, A03d/e, A03x-eghb, A05k, A10a, DLN2, DLN1, A18neo, dsnPMN2, SePN02b, T27Y, TJPMN, Projection neuron, A18c. |
| VL/VO | MSNISNb/d (RP5) | A27h, A03a5, A06l, A06c, A06f, A02g, A02e, A02b, A03a4, A03a6, A03x-eghb, A03d/e, A19d, A27Uniq, DLN1, SePN02b. |

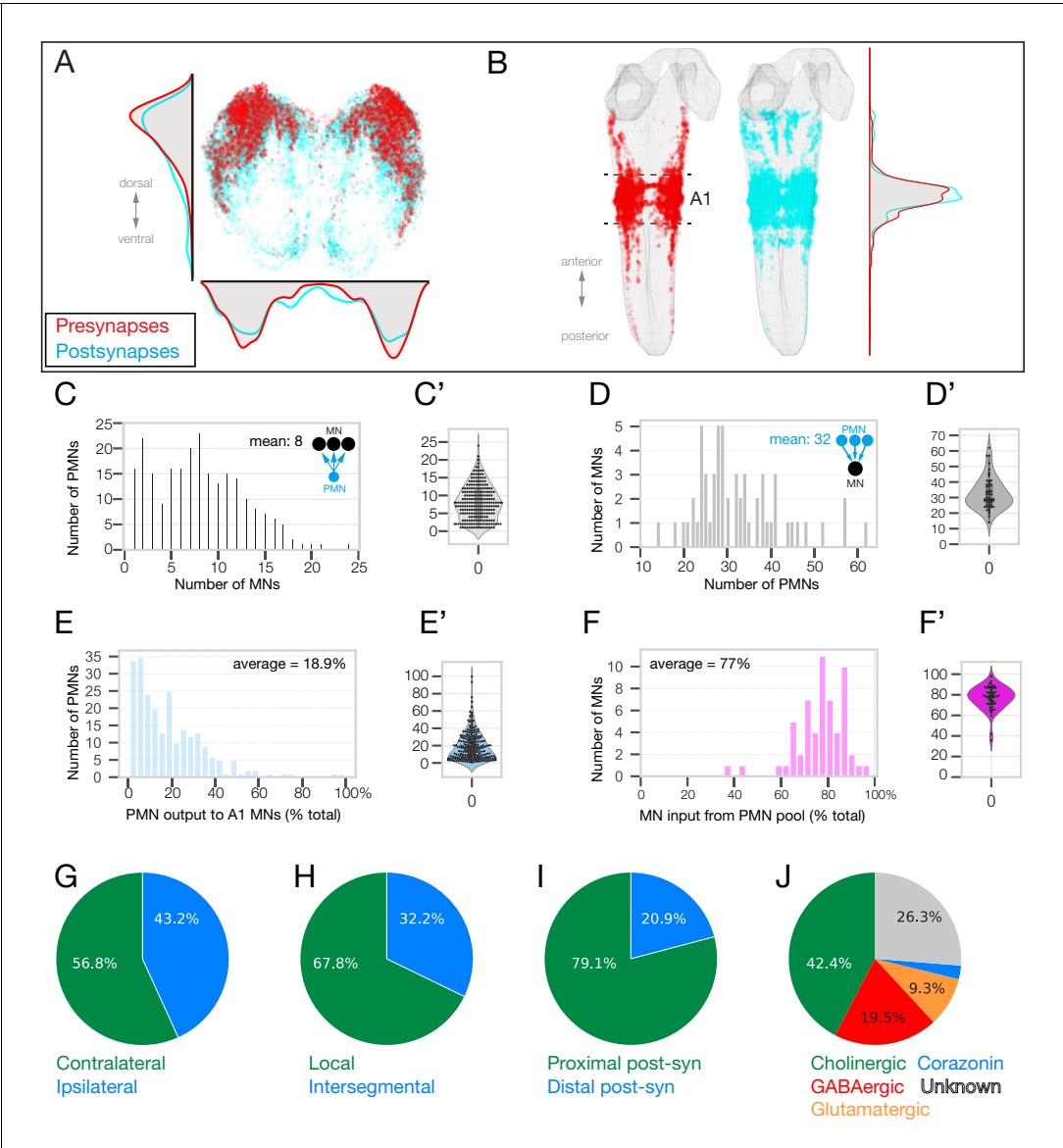

**Figure 6.** Identification of 118 premotor neurons at synapse-level in the TEM volume. (A) Posterior (cross-section) view of the PMN pre-synapse location (red) and postsynapse location (cyan) within the A1 neuropil. Density plots shown for the dorsoventral axis (left) and mediolateral axis (bottom). Dorsal, up. (B) Dorsal view of entire larval neuropil to show anteroposterior distribution of presynapses (red) and postsynapses (cyan). Density plots shown for the anteroposterior axis. (C–F) Quantification of PMN-MN connectivity. All A1 MNs, A2 MN-25, and 118 pair of PMNs were used to generate these histograms. (C) PMNs innervate an average of 8 MNs. X-axis shows binned number of MNs receiving inputs from PMNs. Y-axis shows number of PMNs in each bin (C') Swarm-violin plot representation of the same dataset used in C. (D) MNs receive inputs from an average of 32.5 PMNs from this population of PMNs. X-axis shows binned number of PMNs providing output to MNs. Y-axis shows number of MNs in each bin. (D') Swarm-violin plot representation of the same dataset used in D. (E) Histogram showing binned fraction of PMN output to MNs. Y-axis shows number of PMNs in each bin. (E') Swarm-violin plot representation of the same dataset used in E. (F) Histogram showing binned fraction of MN inputs from PMNs. Y-axis shows number of MNs in each bin. (F') Swarm-violin plot representation of the same dataset used in F. (G–J) Quantification of PMN morphology and neurotransmitter expression. We did not assay Corazonin+ neurons for fast neurotransmitter expression, but a recent RNAseq study shows promiscuous expression of fast neurotransmitters in Corazonin+ neurons (*Brunet Avalos et al., 2019*).

The online version of this article includes the following figure supplement(s) for figure 6:

**Figure supplement 1.** Pre-motor neuron/motor neuron synapse identification in the TEM volume.
**Figure supplement 2.** All premotor neurons traced in the TEM volume.

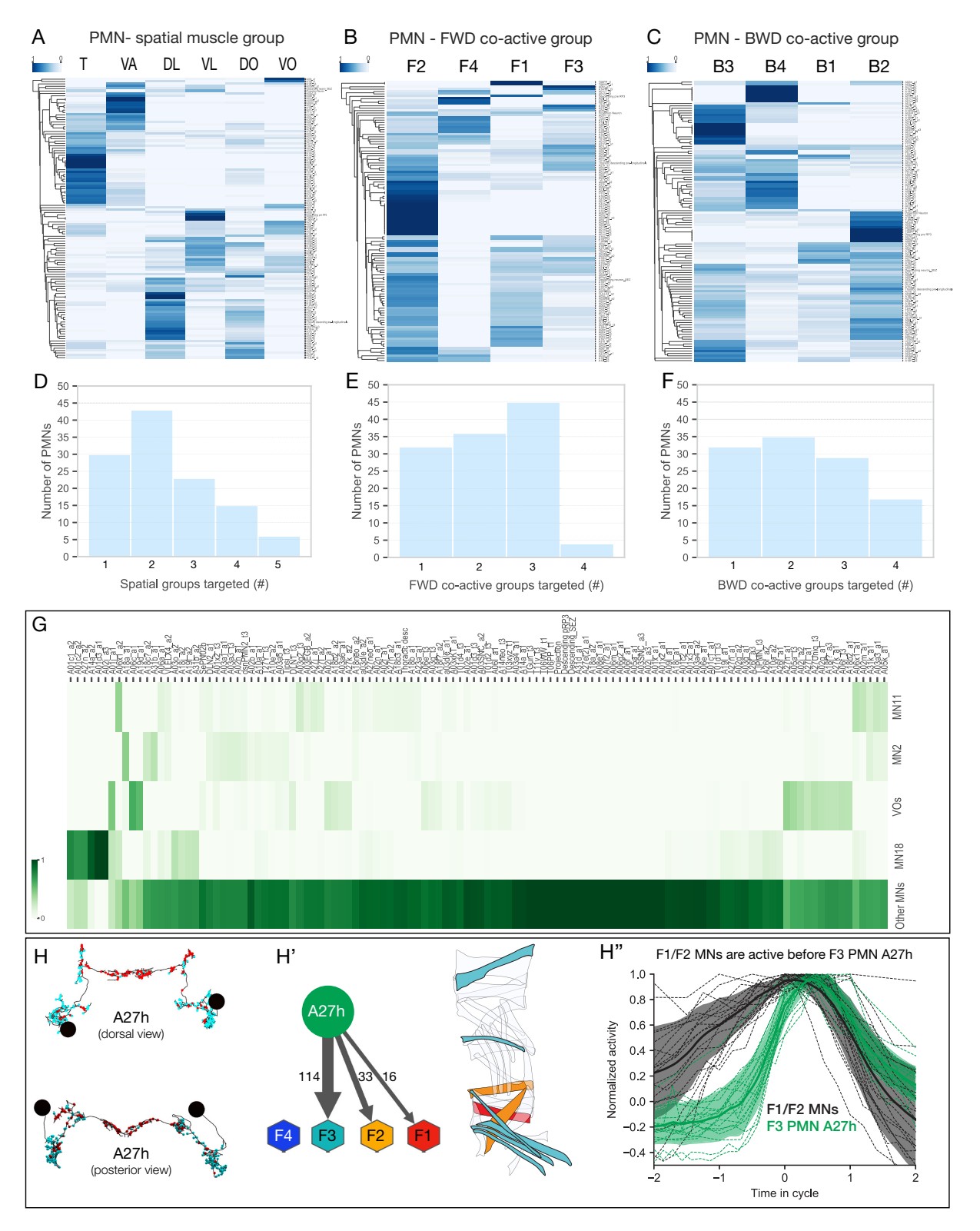

**Figure 7.** PMN pools preferentially connected to individual spatial muscle groups and co-activated muscle groups. (A–C) Hierarchical clustering of PMNs based on their connectivity to type Ib MNs of the same spatial muscle group (A), forward co-activated muscle group (B), or backward co-activated muscle group (C). Heat maps represent the sum of normalized weighted-synaptic output of a given left/right pair of PMNs onto left/right pair of MNs grouped in each panel. Values in each row were normalized to sum to 1. (D–F) Quantification of connectivity between PMNs and type Ib MNs

*Figure 7 continued on next page*

*Figure 7 continued*

innervating spatial muscles (D), forward co-active (E), and backward co-active groups. PMN-MN connections with total weighted synapses of less than 1% were excluded from these analyses. (D) X-axis shows binned number of spatial muscle groups which receive inputs from PMNs. Y-axis shows number of PMNs in each bin. While 30 PMNs connect to only one spatial muscle group, the rest of the PMNs connect to more than one groups. (E) X-axis shows binned number of forward co-active groups which receive inputs from PMNs. Y-axis shows number of PMNs in each bin. While 32 PMNs connect to only one forward co-active group, the rest of the PMNs connect to more than one groups. (F) X-axis shows binned number of backward co-active groups which receive inputs from PMNs. Y-axis shows number of PMNs in each bin. While 32 PMNs connect to only one backward co-active group, the rest of the PMNs connect to more than one groups. (G) Connectivity pattern of PMNs to differentially recruited MNs (11, 2, 18, and VOs) versus other type Ib MNs (Other MNs). Heat maps represent the sum of normalized weighted-synaptic output of a given left/right pair of PMNs onto left/right pair of MNs in each group. Values in each column were normalized to sum to 1. (H–H'') A27h is active following MNs in forward co-active groups F1/F2. (H) Morphology of the reconstructed A27h in segment A1 left and right, showing presynapses (red) and postsynapses (cyan). (H') Pattern of A27h connectivity showing preferential output to MNs active in co-active group F3. Hexagons represent MNs preferentially active in the F1-F4 co-active groups. Muscles innervated by MNs targeted by A27h are shown. (H'') Dual color calcium imaging of jRCaMP1b in A27h (green) and GCaMP6m in U1-U5 MNs (black: MN2, MN3, MN4, MN9, MN10). Consistent with predictions from the connectome, U1-U5 MNs (co-activated muscle group F1/2) are activated before A27h (co-activated muscle group F3). Green and dark error bars (ribbons) represent the standard deviation of the average neuronal activity. Genotype: *CQ-lexA/+; lexAop-GCaMP6m/R36G02-Gal4 UAS-jRCaMP1b*.

The online version of this article includes the following figure supplement(s) for figure 7:

**Figure supplement 1.** Morphology and connectivity of premotor neurons innervating one or more co-active motor neurons (F1–F4).

is active during backward locomotion (*Figure 9C*). Experimental data show that A18a and A18b3

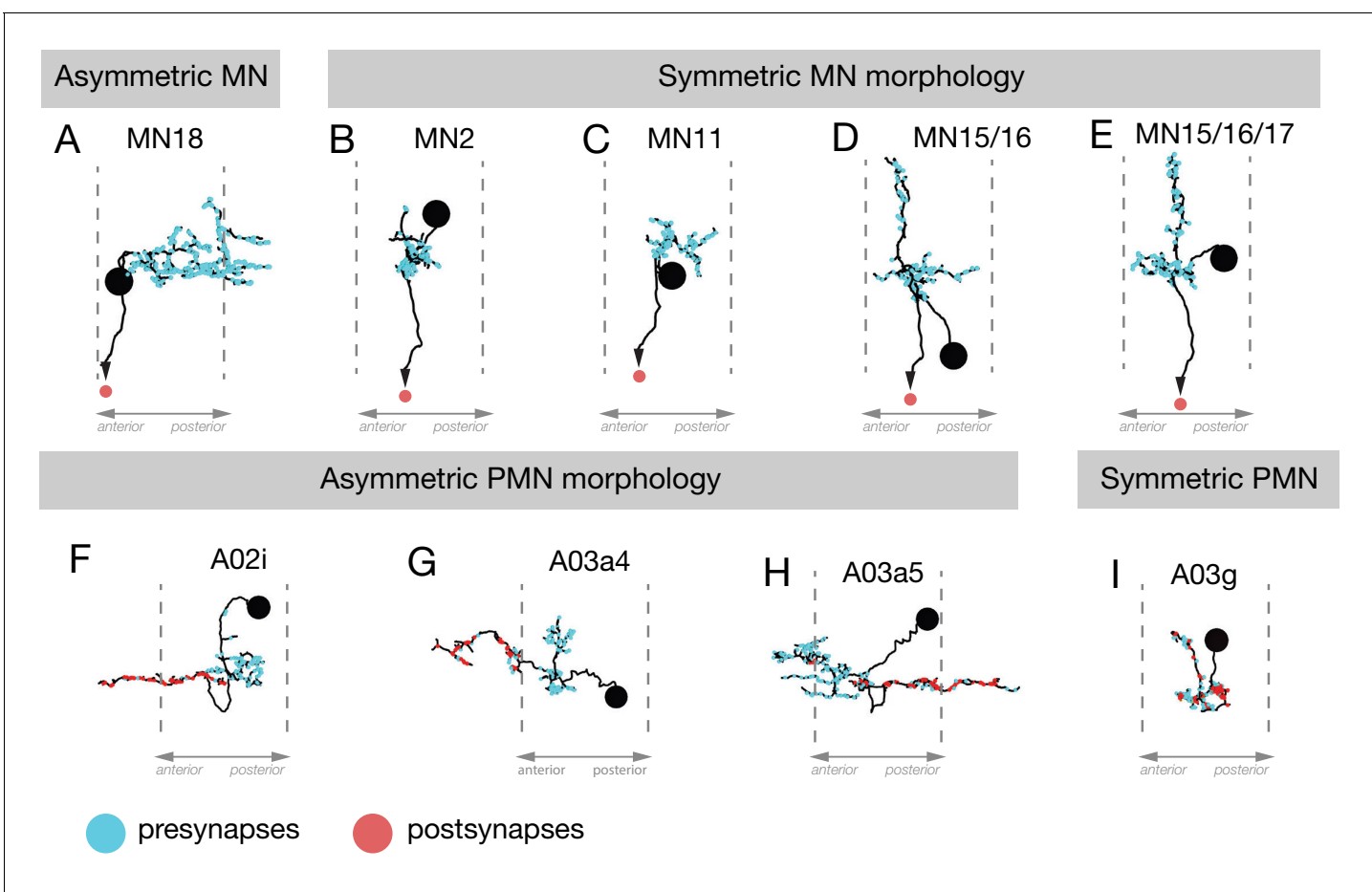

**Figure 8.** Neuronal asymmetry along the anterior-posterior axis. (A–E) MN18 has asymmetric dendrites extending to the next posterior segment, but the dendritic arbors of other differentially recruited MNs (2, 11, 15/16, 15/16/17) were not asymmetric along this axis. (F–I) The PMNs A02i, A03a4 and A03a5 have asymmetric dendrite projections to the anterior (F,G) or posterior (H) of their cell body and presynaptic domain, whereas A03g is an example of a PMN that is symmetric along the anteroposterior axis.

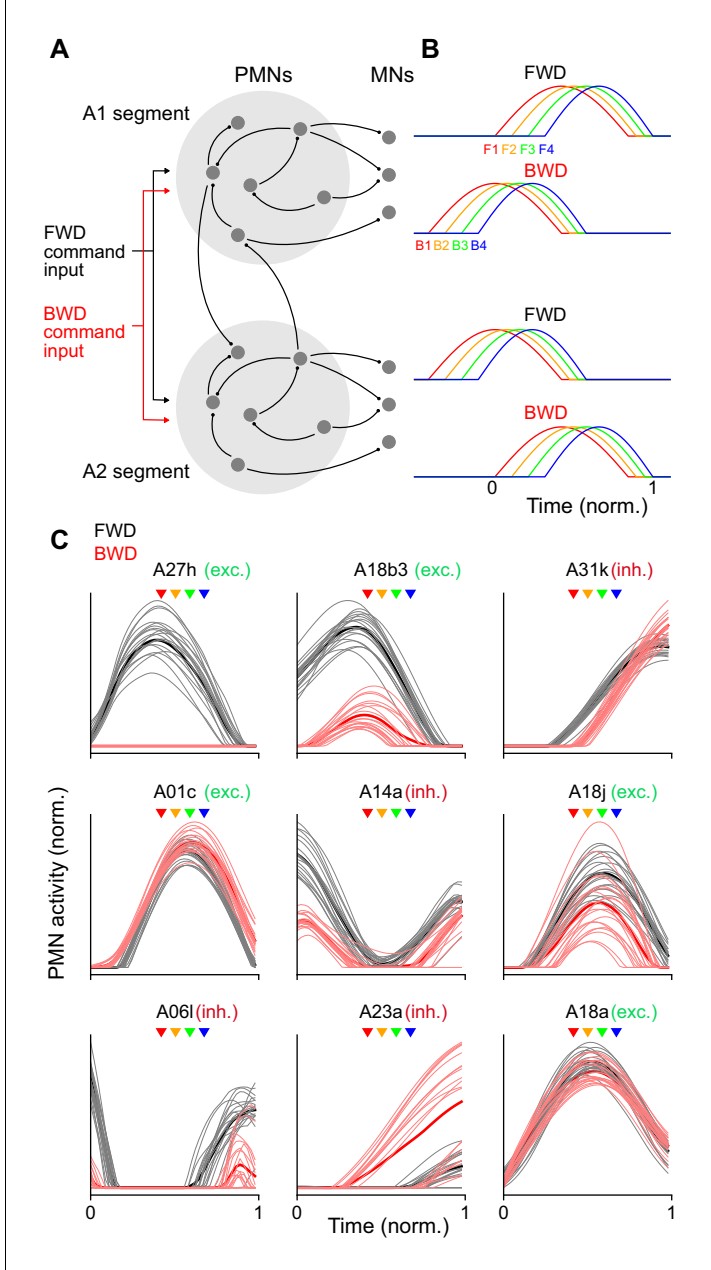

**Figure 9.** Recurrent network model generating sequential MN activity. (**A**) The PMN and MN network of the A1 and A2 segments was modeled using connectivity taken from the TEM reconstruction. Connections within each segment (light gray circles) are identical. The network was optimized using gradient descent to produce a sequential pattern of activity in the MNs when a tonic external command input for forward (FWD, black) or backward (BWD, red) locomotion was applied. (**B**) The network in A was optimized to produce an appropriate sequential activity pattern of co-activated muscle groups during forward and backward crawling. The direction of propagation from the posterior (A2) to anterior (A1) segment or vice versa differs for forward and backward crawling. To compare PMN activity relative to MN activation, time is measured in units normalized to the onset and offset of MN activity in a segment (bottom right). (**C**) Y-Axis is the normalized activity of a subset of PMNs in the model during forward and backward crawling. Thick lines denote averages over the ensemble of models generated. X-axis (time) is measured relative to A1 MN onset and offset as in B. Arrowheads denote the peak activation onset time for the MNs innervating different co-activated muscle groups (color key as in panel B); exc, excitatory; inh, inhibitory.

The online version of this article includes the following figure supplement(s) for figure 9:

*Figure 9 continued on next page*

*Figure 9 continued*

**Figure supplement 1.** Recurrent network model of PMNs activity aligned to onset and offset of A1 MNs during locomotion.
**Figure supplement 2.** Models constructed without constraints on the activity of A27h/A18b.

are active precisely as proposed in our model (*Hasegawa et al., 2016*). Furthermore, our model predicts the cholinergic A18j and A01c PMNs are active at F4, which is supported by experimental data on these neurons (where they were called eIN1,2; *Zwart et al., 2016*).

To provide new, additional experimental tests of our model, we performed dual color calcium imaging on previously uncharacterized GABAergic PMNs A31k and A06l. Our model predicted that both A31k and A06l neurons show peak activity later than the early-activated MNs during both forward and backward locomotion (*Figure 9C*; *Figure 9—figure supplement 1*). To determine experimentally the phase-relationship between A31k and MNs, we expressed GCaMP6m in a subset of MNs and jRCaMP1b in A31k. Dual color calcium imaging data revealed that the A31k activity peak coincides with a decline of activity in MNs innervating early co-activated muscle groups during both forward and backward locomotion (*Figure 10A,B*), further validating our model. Second, our model predicts that both A31k and A06l PMNs show concurrent, rhythmic activity during forward and backward locomotion (*Figure 9—figure supplement 1*). We expressed GCaMP6m in both neurons, which we could distinguish based on their different axon projections, and found that they showed concurrent, rhythmic activity (*Figure 10C,D*), and thus both neurons show a delayed activation relative to MNs. Our third experimental test focused on the GABAergic A23a PMN (*Schneider-Mizell et al., 2016*). Our model predicts that A23a was active earlier during backward locomotion than forward locomotion (*Figure 9C*). We expressed GCaMP6m in a subset of MNs and jRCaMP1b in A23a, and validated the prediction of our model (*Figure 10E,F*).

We conclude that our model accurately predicts many, but not all, of the experimentally determined PMN-MN phase relationships (see Discussion). With the exception of *C. elegans* models (*Karbowski et al., 2008*; *Macosko et al., 2009*; *Wen et al., 2012*; *Izquierdo and Beer, 2013*; *Izquierdo et al., 2015*; *Kunert et al., 2017*; *Rakowski and Karbowski, 2017*), the networks constructed here represent the first models of the neural circuitry underlying a motor behavior whose connectivity has been constrained by a synaptic wiring diagram. Prior studies of *C. elegans* have highlighted the importance of proprioception in order to drive locomotion (*Kunert et al., 2017*), while our model does not require proprioceptive input to generate the observed motor pattern, consistent with data showing that an isolated CNS without sensory input (including no proprioception) can maintain forward and backward waves of motor neuron activity (*Pulver et al., 2015*). Our study also includes stronger constraints on excitatory and inhibitory signaling, which are difficult to infer in *C. elegans* (*Rakowski and Karbowski, 2017*).

## Circuit motifs specific for forward or backward locomotion

PMNs, in addition to connecting to MNs, make presynapses onto other neurons (*Supplementary file 6*), generating circuit motifs that may play important roles during larval locomotion. Interestingly, some of these PMNs are active only during forward or backward locomotion (*Fushiki et al., 2016*; *Carreira-Rosario et al., 2018*; *Kohsaka et al., 2019*), indicating they may change the dynamics of motor circuits during forward versus backward locomotion, resulting in different muscle activity patterns during forward or backward crawling. Here we used connectome and neurotransmitter data to examine circuit motifs that include these direction-specific PMNs and asked how they can contribute to the generation of different coactive muscle groups during forward and backward locomotion.

The previously described forward-specific excitatory PMN A27h (*Fushiki et al., 2016*; *Carreira-Rosario et al., 2018*), with F3 onset, connects to another forward-specific excitatory PMN A18b3 (*Hasegawa et al., 2016*) innervating F1-F2 MNs. Thus, when A27h activates F3, it also maintains activity of A18b3 to ensure continued excitation of F1/F2 MNs (*Figure 11A*). These motifs provide testable hypotheses for how specific phase relationships between co-activated muscle groups are generated by PMNs. A27h also excites PMN A18b3 in the next anterior segment, which could advance the intersegmental forward contraction wave; similarly, A18b3 excites two inhibitory PMNs

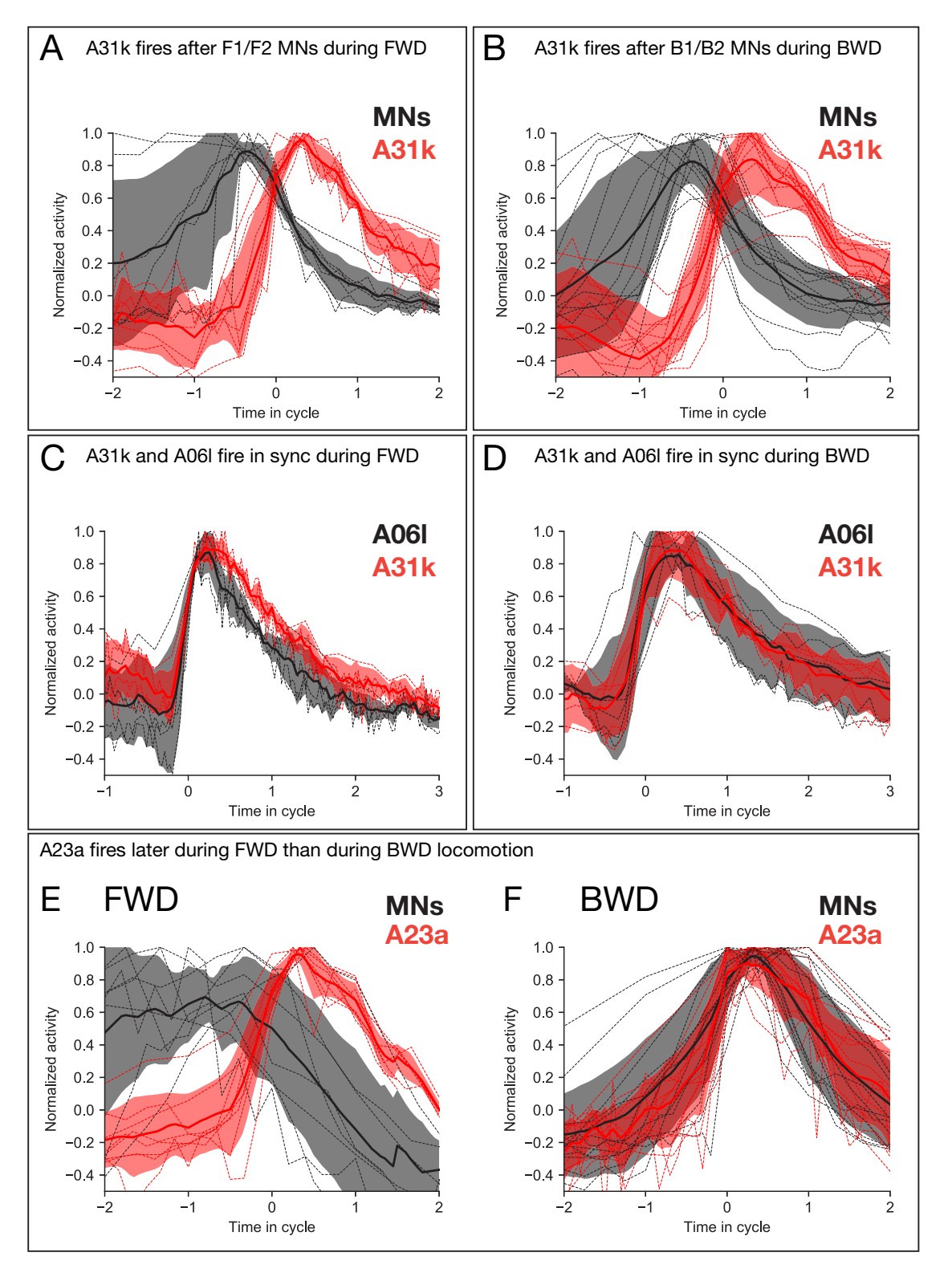

**Figure 10.** Calcium imaging of A31k/A06l/A23a PMNs and their target MNs validates the activity pattern predicted by recurrent modeling. (**A–B**) Dual color calcium imaging of jRCaMP1b in A31k (red) and GCaMP6m in MNs (black). Consistent with the recurrent model predictions, A31k fires with a delay after its postsynaptic MNs in both forward (**A**) and backward (**B**) waves. Red and dark error bars (ribbons) represent the standard deviation of the average neuronal activity. Genotype: *CQ-lexA/+; lexAop-GCaMP6m/R87H09-Gal4 UAS-jRCaMP1b*. (**C–D**) Single color calcium imaging of jRCaMP1b in

*Figure 10 continued on next page*

*Figure 10 continued*

A31k (red) and A06l (black). Consistent with the recurrent model predictions, A31k and A06l show synchronous activity patterns during forward (C) and backward waves (D). Red and dark error bars (ribbons) represent the standard deviation of the average neuronal activity. Genotype: *R87H09-Gal4 UAS-jRCaMP1b*. (E,F) A23a fires later during forward locomotion than during backward locomotion. Dual color calcium imaging of jRCaMP1b in A23a (red) and GCaMP6m in MNs (black). Red and dark error bars (ribbons) represent the standard deviation of the average neuronal activity. Genotype: *CQ-lexA/+; lexAop-GCaMP6m/R78F07-Gal4 UAS-jRCaMP1b*.

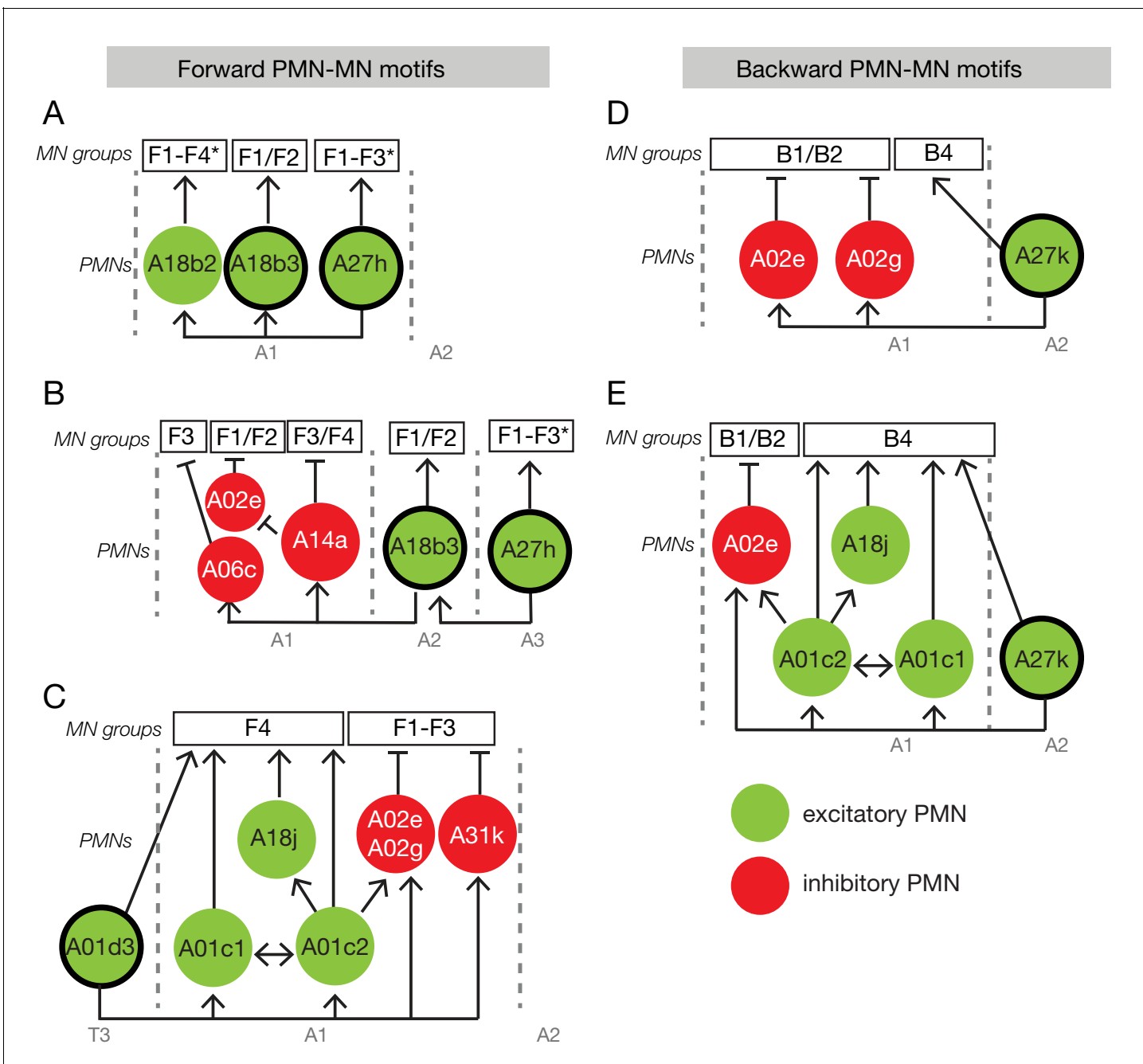

**Figure 11.** Neural circuit motifs specific for forward or backward locomotion. Circuit motifs composed of forward-specific PMNs (A–C) and backward specific PMNs (D–E). See text for details. Arrow/green, excitatory connection; T-bar/red, inhibitory connection; F1-F4, forward co-active group; B1-B4, backward co-active group. Thick outlines indicate forward-specific or backward-specific PMNs.

(A06c, A14a) in the next anterior segment which may prevent premature activation of F3/4 MNs (*Figure 11B*). Moreover, consistent with F1/F2 vs F3/F4 temporal segregation, A14a disinhibits F1/F2 MNs via silencing A02e PMN (*Figure 11B*). Another forward-specific PMN (A01d3; also known as ifb-FWD, *Kohsaka et al., 2019*), is also a component of feedforward excitation and feedforward inhibition motifs that may temporally segregate F1-F3 from F4 coactive-muscle groups (*Figure 11C*).

Next, we examined circuit motifs composed of a backward-specific PMN, A27k (also known as ifb-BWD) (*Kohsaka et al., 2019*). A27k excites B4 MNs as well as the inhibitory PMNs A02e and A02g innervating B1/B2. This motif could coordinate excitation of B3/B4 MNs and termination of B1/B2 MN activity as the contraction wave moves posteriorly (*Figure 11D*). A27k also synapses in the next anterior segment with the excitatory neurons A01c1, A01c2, and A18j (innervating B4), as well as with the inhibitory PMN A02e innervating B1/B2. This could coordinately terminate B1/B2 MN activity and activate B4 MN activity (*Figure 11E*). Thus, we identified both feedforward excitation and feedforward inhibition motifs that could explain the sequential activation of a specific coactivated muscle group in adjacent segments during backward motor waves. We conclude that circuit motifs composed of forward or backward specific PMNs are likely to be an additional mechanism for generating distinct forward or backward coactivated muscle groups. Functional examination of these motifs is beyond the scope of the current study.

## Discussion

It is a major goal of neuroscience to comprehensively reconstruct neuronal circuits that generate specific behaviors, but to date this has been done only in *C. elegans* (*Karbowski et al., 2008*; *Macosko et al., 2009*; *Izquierdo and Beer, 2013*; *Izquierdo et al., 2015*; *Kunert et al., 2017*; *Rakowski and Karbowski, 2017*). Recent studies in mice and zebrafish have shed light on the overall distribution of PMNs and their connections to several well-defined MN pools (*Eklöf-Ljunggren et al., 2012*; *Kimura et al., 2013*; *Bagnall and McLean, 2014*; *Ljunggren et al., 2014*). However, in mouse and zebrafish it remains unknown if there are PMNs that have yet to be characterized, and the connectivity between PMNs is not well described, which would be important for understanding the network properties that produce coordinated motor output. In the locomotor central pattern generator circuitry of leech, lamprey, and crayfish, the synaptic connectivity between PMNs or between PMNs and other interneurons is known to play critical roles in regulating the swimming behavior (*Brodfuehrer and Thorogood, 2001*; *Grillner, 2003*; *Kristan et al., 2005*; *Mullins et al., 2011*; *Mulloney and Smarandache-Wellmann, 2012*; *Mulloney et al., 2014*). However, it is difficult to be certain that all the neural components and connections of these circuits have been identified. The comprehensive anatomical circuitry reconstructed in our study provides an anatomical constraint on the functional connectivity used to drive larval locomotion; all synaptically-connected neurons may not be relevant, but at least no highly connected local PMNs are absent from our analysis.

Our results extend previous studies of *Drosophila* larval locomotion. Previous work has suggested that several muscles recruited at different times during the crawl cycle receive input from distinct populations of premotor neurons. For example, a recent study (*Zwart et al., 2016*) has shown that the GABAergic A14a inhibitory PMN (also called iIN1) selectively inhibits MNs innervating muscle 22/LT2 (co-activated muscle group F4), thereby delaying muscle contraction relative to muscle 5/LO1 (co-activated muscle group F2). While this model suggests a labeled-line mechanism, we also find that A14a disinhibits MNs in early co-activated muscle groups F1/2 via the inhibitory PMN A02e. Thus, A14a both inhibits late co-activated muscle groups and disinhibits early co-activated muscle groups. In addition to A14a, we find that the majority of PMNs target populations of MNs that span more than one co-active muscle group suggesting that sequential recruitment may primarily arise from a combinatorial network of PMNs. In addition, previous work has suggested that all MNs receive simultaneous excitatory inputs from different cholinergic PMNs (*Zwart et al., 2016*). However, our dual calcium imaging data show that the excitatory A27h PMN is active during the F3 co-activated muscle group and not earlier. Therefore, MNs may receive temporally distinct excitatory inputs, in addition to the previously reported temporally distinct inhibitory inputs. We have identified dozens of new PMNs that are candidates for regulating motor rhythms; functional analysis of all of these PMNs is beyond the scope of this paper, particularly due to the additional work required to

screen and identify Gal4/LexA lines selectively targeting these PMNs, but our predictions are clear and testable when reagents become available.

We show that MNs innervating a single spatial muscle group can belong to more than one co-activated muscle group, therefore spatial muscle groups do not invariably match co-activated muscle groups. This could be due to several reasons: (i) MNs in each spatial muscle groups receive inputs from overlapping but not identical array of PMNs. (ii) Different MNs in the same spatial muscle group receive a different number of synapses from the same PMN. (iii) MNs in the same spatial muscle group vary in overall dendritic size and total number of postsynapses, thereby resulting in MNs of the same spatial muscle group falling into different co-activated muscle groups.

We demonstrate that during both forward and backward locomotion, most of longitudinal and transverse muscles of a given segment contract as early and late groups, respectively. In contrast, muscles with oblique or acute orientation often show different phase relationships during forward and backward crawling. Future studies will be needed to provide a biomechanical explanation for why oblique muscles – but not longitudinal or transverse muscles – need to be recruited differentially during forward or backward crawling. Also, it will be interesting to determine whether the VO or VL MNs are responsible for elevating the ventral cuticular denticles during propagation of the peristaltic wave; if the VOs, it would mean that lifting the denticles occurs at different phases of the crawl cycle in forward and backward locomotion.

Our recurrent network model accurately predicts the order of activation of specific PMNs, yet many of its parameters remain unconstrained, and some PMNs may have biological activity inconsistent with activity predicted by our model. Sources of uncertainty in the model include incomplete reconstruction of inter-segmental connectivity and descending command inputs, the potential role of gap junctions (which are not resolved in the TEM reconstruction), as well as incomplete characterization of PMN and MN biophysical properties. Recent studies have suggested that models constrained by TEM reconstructions of neuronal connectivity are capable of predicting features of neuronal activity and function in the *Drosophila* olfactory (*Eichler et al., 2017*) and visual (*Takemura et al., 2013*; *Tschopp et al., 2018*) systems, despite the unavoidable uncertainty in some model parameters and the likely presence of multiple distinct models that produce activity consistent with recordings (*Prinz et al., 2004*; *Brenner, 2010*; *Bargmann and Marder, 2013*). For the locomotor circuit described here, we anticipate that the addition of model constraints from future experiments will lead to progressively more accurate models of PMN and MN dynamics. Despite its limitations, the ability of the PMN network to generate appropriate muscle timing for two distinct behaviors in the absence of third-layer or command-like interneurons suggests that a single layer of recurrent circuitry is sufficient to generate multiple behavioral outputs. It is also notable that a model lacking complex single-neuron dynamics such as post-inhibitory rebound or spike-frequency adaptation, which are critical for modeling other central pattern generator circuits (*Marder and Bucher, 2001*), is sufficient to produce the observed motor pattern. Thus, although there are likely complex intrinsic neuronal dynamics that our model fails to capture, recurrent excitatory and inhibitory interactions may play a large role in establishing appropriate motor timing in the larva.

Previous work in other animal models have identified multifunctional muscles involved in more than one motor behavior: swimming and crawling in *C. elegans* (*Pierce-Shimomura et al., 2008*; *Vidal-Gadea et al., 2011*; *Butler et al., 2015*) and leech (*Briggman and Kristan, 2006*); walking and flight in locust (*Ramirez and Pearson, 1988*); respiratory and non-respiratory functions of mammalian diaphragm muscle (*Lieske et al., 2000*; *Fogarty et al., 2018*) unifunctional muscles which are only active in one specific behavior in the lobster *Homarus americanus* (*Mulloney et al., 2014*); swimming in the marine mollusk *Tritonia diomedea* (*Popescu and Frost, 2002*); and muscles in different regions of crab and lobster stomach (*Bucher et al., 2006*; *Briggman and Kristan, 2008*). Our single-muscle calcium imaging data indicate that all imaged larval body wall muscles are bifunctional and are activated during both forward and backward locomotion. It will be interesting to determine if all bodywall muscles are also involved in other larval behaviors, such as escape rolling, self-righting, turning, or digging. It is likely that there are different co-activated muscle groups for each behavior, as we have seen for forward and backward locomotion, raising the question of how different co-activated muscle groups are generated for each distinct behavior.

# Materials and methods

## Key resources table

| Reagent type (species) or resource | Designation | Source or reference | Identifiers | Additional information |
|---|---|---|---|---|
| Genetic reagent (*Drosophila melanogaster*) | R36G02-Gal4 | BDSC | # 49939 | A27h line |
| Genetic reagent (*Drosophila melanogaster*) | R87H09-Gal4 | BDSC | #40507 | A31k and A06l line |
| Genetic reagent (*Drosophila melanogaster*) | R78F07-Gal4 | BDSC | #47409 | A23a line |
| Genetic reagent (*Drosophila melanogaster*) | R44H10-LexA | BDSC | # 61543 | Muscle line |
| Genetic reagent (*Drosophila melanogaster*) | CQ2-lexA | this paper | | U1-U5 motor neurons |
| Genetic reagent (*Drosophila melanogaster*) | UAS-jRCaMP1b | BDSC | # 63793 | calcium indicator |
| Genetic reagent (*Drosophila melanogaster*) | lexAop-GCaMP6m | BDSC | #44275 | calcium indicator |
| Genetic reagent (*Drosophila melanogaster*) | 13XLexAop2-6XmCherry-HA | BDSC | #52271 | Used for ratiometric muscle calcium imaging |

## Electron microscopy and CATMAID reconstructions

The TEM volume is for an L1 newly hatched larva (*Ohyama et al., 2015*), and is available by request from Albert Cardona (Cambridge University). Neurons were reconstructed in CATMAID (*Saalfeld et al., 2009*) using a Google Chrome browser as previously described (*Ohyama et al., 2015*). Figures were generated using CATMAID graph or 3D widgets combined with Adobe Illustrator (Adobe, San Jose, CA).

## Criteria for selecting PMNs

The 118 PMNs were selected if they had presynapses to greater than 1% of a MN postsynapse population and greater than four total synapses (left+right combined). Candidate PMNs that met these criteria could be excluded however if left/right orthologs could not be identified, or if gaps prevented reliable reconstruction of the PMN in the neuropil regions of T3, A1, and A2 segments.

## Synapse spatial distributions and clustering

Synapse spatial distributions were generated using custom MATLAB scripts, which are deposited at GitHub (*Mark and Litwin-Kumar, 2019*). Spatial distributions were determined using kernel density estimates with a 1 µm bandwidth. For cross-sectional spatial distributions, points were rotated −12 degrees around the Z-axis (A/P axis) in order to account for the slight offset of the EM-volume. For pre-synaptic sites, polyadic synapses were weighted by their number of postsynaptic targets. Synapse similarity was calculated as described previously (*Schlegel et al., 2016*):

$$f(is,jk) = e^{\frac{-d_{sk}^2}{2\sigma^2}} e^{-\frac{|n_{is}-n_{jk}|}{n_{is}+n_{jk}}}$$

where *f(is,jk)* is the mean synapse similarity between all synapses of neuron *i* and neuron *j*. $d_{sk}$ is the Euclidean distance between synapses *s* and *k* such that synapse *k* is the closest synapse of neuron *j* to synapse s of neuron *i*. σ is a bandwidth term that determines what is considered close. $n_{is}$ and $n_{jk}$ are the fraction of synapses for neuron *i* and neuron *j* that are within ω of synapse *s* and synapse *k* respectively. For MN inputs, σ = ω = 2 µm. Clustering was performed by using the average synapse similarity scores for the left and right hemisegments as a distance metric, and linkage was calculated using the average synapse similarity. For comparing the distributions across individual axes, a two sample Kolmogorov-Smirnov test was used to determine significance.

## Clustering analysis of PMN-MN connectivity

Weighted PMNs to MNs connectivity matrix was acquired from CATMAID TEM volume as percentage of total number of postsynaptic links to these target MNs. We then calculated the average of left and right pairs of PMNs and MNs. Next, we summed the mean connections from PMNs to all MNs innervating muscle groups defined in *Figure 7A–C and G*, and normalized the values for each row (PMNs). Hierarchical clustering was performed on these normalized connectivity matrixes using Python's seaborn.clustermap (metric = Euclidean, method = single, https://seaborn.pydata.org/generated/seaborn.clustermap.html). For *Figure 7D–F*, the PMN-MN connectivity matrix was acquired from CATMAID as the absolute number of presynapses from PMNs to their target MNs. We first summed the weighted synapses from left and right counterparts of individual PMNs to individual MNs. Then we calculated the average of these values for MN pairs (left and right) and summed the connections from PMNs to all MNs innervating the spatial or co-active muscle groups. To generate 7D-F, we discarded PMN-MN connections where total weighted synapses between a given PMN pair and MN pair were less than 1%. Histograms were produced in Python sns.distplot (https://seaborn.pydata.org/generated/seaborn.distplot.html).

## Muscle GCaMP6f imaging, length measurement, and quantification

2% melted agarose was used to make pads with similar size: 25 mm (W) X 50 mm (L) X 2 mm (H). Using tungsten wire, a shallow ditch was made on agarose pads to accommodate the larva. To do muscle ratiometric calcium imaging in intact animals, a first or second instar larvae expressing GCaMP6f and mCherry in body wall muscles were washed with distilled water, then moved into a 2% agarose pad on the slide. A 22 mm ×40 mm cover glass was put on the larva and pressed gently to gently constrain larval locomotion. The larva was mounted dorsolaterally or ventrolaterally to image a different set of muscles (dorsolateral mount excludes the most ventral muscles (15,16,17) whereas the ventrolateral mount excludes the dorsal-most muscles (1,2,9,10); imaging was done with a 10x objective on an upright Zeiss LSM800 microscope. We recorded a total of 38 waves (24 forward and 14 backward) from four different animals, and examined muscle calcium activity in two subsequent hemi-segments for each wave. Muscle length measurement was done using custom MATLAB scripts where muscle length was measured on a frame by frame basis. Calcium imaging data were also analyzed using custom MATLAB scripts. Due to movement artifacts, ROIs were updated on a frame by frame basis to track the muscle movement. ROIs that crossed other muscles during contraction were discarded. In no single preparation was it possible to obtain calcium traces for all 30 muscles. Instead, we used only preparations in which at least 40% of the muscles could be recorded. In order to align crawl cycles that were of variable time and muscle composition, we first produced a two dimensional representation of each crawl cycle using PCA. Crawl cycles were represented as circular trajectories away from, and back towards the origin (*Figure 3—figure supplement 1E,F*) similar to what has been shown previously (*Lemon et al., 2015*). The amplitude, or linear distance from the origin, to a point on this trajectory correlated well with both the coherence of the calcium signals as well as the amplitude of the population. Thus, peaks in this 2D amplitude correspond with the time in which most muscles are maximally active, which we defined as the midpoint of a crawl cycle. It should be noted that the muscles used to generate two dimensional representations of crawl cycles were different for each crawl. While this means that each PCA trajectory is slightly different for each crawl cycle, we reasoned that because each experiment contained muscles from every co-activated muscle group, the peak amplitude in PCA space should still correspond to a good approximation of the midpoint of the crawl cycle. We defined the width of a crawl cycle as the width of this 2D peak at half-height (*Figure 3—figure supplement 1G*). We aligned all crawl cycles to the crawl onset and offset (which we call 25% and 75% of the crawl cycle respectively) as defined by this width at half-height (*Figure 3—figure supplement 1H,I*).

## Calcium imaging in neurons

For dual-color and single-color calcium imaging in fictive preps, freshly dissected brains were mounted on 12 mm round Poly-D-Lysine Coverslips (Corning BioCoat) in HL3.1 saline (*de Castro et al., 2014*), which were then were placed on 25 mm ×75 mm glass slides to be imaged with a 40 × objective on an upright Zeiss LSM-800 confocal microscopy. To simultaneously image two different neurons expressing GCaMP6m we imaged neuron-specific regions of interest (ROI). In

addition, we imaged two neurons differentially expressing GCaMP6m and jRCaMP1b. Image data were imported into Fiji (https://imagej.net/fiji) and GCaMP6m and jRCaMP1b channels were separated. The $\Delta F/F_0$ of each ROI was calculated as $(F-F_0)/F_0$, where $F_0$ was averaged over ~1 s immediately before the start of the forward or backward waves in each ROI.

### Neurotransmitter expression

For PMNs that we could identify using Gal4 lines, we crossed the Gal4 line to UAS-Cherry and used Chat:GFP to detect cholinergic neurons (*Diao et al., 2015*), anti-GABA to detect GABAergic neurons (*Wilson and Laurent, 2005*), anti-vesicular glutamate transporter (vGlut) to detect glutamatergic neurons (*Daniels et al., 2004*), or anti-Corazonin to detect corazonergic neurons (*Veenstra and Davis, 1993*).

### Antibody staining and imaging

Standard confocal microscopy, immunocytochemistry and MCFO methods were performed as previously described for larvae (*Carreira-Rosario et al., 2018*). Primary antibodies used: GFP or Venus (rabbit, 1:500, ThermoFisher, Waltham, MA; chicken 1:1000, Abcam13970, Eugene, OR), GFP or Citrine (Camelid sdAB direct labeled with AbberiorStar635P, 1:1000, NanoTab Biotech., Gottingen, Germany), GABA (rabbit, 1:200, Sigma, St. Louis, MO), vGlut (rabbit, 1:10000, gift from Aaron DiAntonio), Corazonin (rabbit, 1:1000), mCherry (rabbit, 1:1000, Novus, Littleton, CO), HA (mouse, 1:200, Cell Signaling, Danvers, MA), V5 (rabbit, 1:400, Rockland, Atlanta, GA), Flag (rabbit, 1:200, Rockland, Atlanta, GA). Secondary antibodies were from Jackson Immunoresearch (West Grove, PA) and used according to manufacturer's instructions. Confocal image stacks were acquired on Zeiss 710 or 800 microscopes. Images were processed in Fiji (https://imagej.net/Fiji), Photoshop, and Illustrator (Adobe, San Jose, CA). Brightness and contrast adjustments were applied to the entire image uniformly; mosaic images were assembled in Photoshop (Adobe, San Jose, CA).

### Recurrent network model

#### Model dynamics

We constructed a recurrent network representing the activity of PMNs, which we denote by the vector **p**, and of MNs, which we denote by the vector **m**. The firing rate of PMN or MN $i$ is a rectified-linear function of its input: $p_i(t) = [u_i^p(t)]_+$ or $m_i(t) = [u_i^m(t)]_+$, where $[\cdot]_+$ denotes rectification. The PMN input $\mathbf{u}^p$ follows the differential equation:

$$\tau^p \odot \frac{d\mathbf{u}^p}{dt} = -\mathbf{u}^p(t) + \mathbf{g}^p \odot (\mathbf{J}^p \mathbf{p}(t) + \mathbf{b}^p + \mathbf{I}(t)),$$

where $\tau_i^p$ is the time constant of PMN $i$, $b_i^p$ its baseline excitability, $I_i(t)$ its descending input from other regions, $\odot$ denotes element-wise multiplication, and $\mathbf{J}^p$ is the connectivity matrix among PMNs. We also include a neuron-specific gain term $g_i^p$ which determines how sensitive a PMN is to its inputs (this is required because we fix the scale of $\mathbf{J}$ based on the TEM reconstruction). The descending input to the PMNs $\mathbf{I}(t)$ is represented as a pulse of activity: $\mathbf{I}(t) = \mathbf{I}^{FWD}$ during FWD crawling, $\mathbf{I}(t) = \mathbf{I}^{BWD}$ during BWD crawling, and $\mathbf{I}(t) = 0$ otherwise.

MNs follow similar dynamics:

$$\tau^m \odot \frac{d\mathbf{u}^m}{dt} = -\mathbf{u}^m(t) + \mathbf{g}^m \odot (\mathbf{J}^m \mathbf{p}(t) + \mathbf{b}^m),$$

where $\mathbf{J}^m$ is the connectivity matrix from PMNs to MNs.

To generate PMNs and MNs corresponding to the A2 segment, we duplicated the A1 MNs and the PMNs we reconstructed for which no corresponding neuron in the next anterior segment was reconstructed. This produces a connectivity matrix with an approximate block structure:

$$\mathbf{J}^p = \begin{pmatrix} \mathbf{J}_{11}^p & \mathbf{J}_{12}^p \\ \mathbf{J}_{21}^p & \mathbf{J}_{22}^p \end{pmatrix}, \ \mathbf{J}^m = \begin{pmatrix} \mathbf{J}_{11}^m & \mathbf{J}_{12}^m \\ \mathbf{J}_{21}^m & \mathbf{J}_{22}^m \end{pmatrix},$$

where $\mathbf{J}_{rs}^{p/m}$ represents connections from segment $r$ to segment $s$.

## Target activity

The model parameters ($\mathbf{J}$, $\mathbf{g}$, $\mathbf{b}$, $\tau$, $\mathbf{I}$) are adjusted using gradient descent so that the MN activity $\mathbf{m}$ reproduces target patterns of activity during FWD and BWD crawling. These targets are defined for 6 s trials that contain one sequence of CMUG activation in each of the two segments. Time is discretized into 50 ms bins. At the beginning of each trial, $\mathbf{u}^p$ is initialized with random values from a truncated Gaussian distribution with standard deviation 0.1, and $\mathbf{u}^m$ is initialized to 0. A trial consists of sequential activity in each segment with a 1 s inter-segmental delay (*Figure 9*). Trials begin and end with 1 and 1.5 s of quiescence, respectively. Each MN's target activity is given by a rectified cosine pulse of activity whose start and end times depend on the CMUG to which it belongs. The first CMUG is active for 2 s, and subsequent CMUGs activate with a delay of 0.25 s between each group and end with a delay of 0.125 s between groups. The participation of MNs in CMUGs and the order in which the segments are active during FWD and BWD crawling are inferred from the data (Figure 3).

## Parameter constraints and optimization

Constraints are placed on the model parameters based on knowledge of the circuit. The nonzero elements of $\mathbf{J}^p$ and $\mathbf{J}^m$ are determined from the TEM reconstruction (normalized based on the percent input received by the postsynaptic target), and signs are constrained using neurotransmitter identity when available. If the neurotransmitter identity of a neuron is not known, we initialize the connection to be inhibitory but do not constrain its sign during optimization. Time constants $\tau$ are constrained to be between 50 ms and 1 s (these represent combined membrane and synaptic time constants), and gains $\mathbf{g}$ are constrained to be positive.

At the beginning of optimization, the biases $\mathbf{b}^p$ and $\mathbf{b}^m$ are initialized equal to 0.1 and 0, respectively. Time constants $\tau$ are initialized to 200 ms and gains $\mathbf{g}$ to 1. $\mathbf{I}^{FWD}$ and $\mathbf{I}^{BWD}$ are initialized uniformly between 0.05 and 0.15 for each neuron. To initialize $\mathbf{J}^p$ and $\mathbf{J}^m$, initial connection strengths are taken in proportion to synapse counts from the TEM reconstruction with a scaling factor of $\pm 0.005$ for excitatory/inhibitory connections. Connections within a model segment are taken from the TEM reconstruction of A1, while connections from A1 to A2 or A2 to A1 are taken from the corresponding cross-segmental reconstructions (and are thus likely less complete than the within-segmental connectivity).

The cost function that is optimized consists of a term $C_{targ}$ that penalizes deviations of the MN activities from their targets and three regularization terms to promote realistic solutions. The target term is given by $C_{targ} = \sum_{t,i} w_i ||m_i(t) - m_i^*(t)||^2$, where $m_i^*(t)$ is the target activity for MN $i$ and $w_i$ is a weighting term, proportional to $1/\sqrt{N_{CMUG,i}}$ where $N_{CMUG,i}$ is the number of neurons in the CMUG of neuron $i$ (this scaling ensures the target patterns of CMUGs with few MNs are still reproduced accurately). The first regularization term is given by $C_{A18b,A27h} = 0.05 \cdot \left( \sum_{t \in FWD} |p_{A18}(t)| + \sum_{t \in BWD} |p_{A27}(t)| \right)$, which suppresses the activity of the A18b and A27h neurons for behaviors during which they are known to be quiescent. The second regularization term $C_{seg}$ constrains PMN activity to reflect the timing of segmental activation. It is given by

$$C_{seg} = \alpha_n \sum_{t \in active1} ||\mathbf{p}_1(t) - \mathbf{p}_2(t - t_{delay})||^2,$$

where active1 represents the times when segment 1 is active, $\mathbf{p}_1$ and $\mathbf{p}_2$ represent vectors of PMN activities corresponding to pairs of homologous neurons in adjacent segments, and $t_{delay}$ is the time delay between segment 1 and 2 activations (equal to -1 s for forward and +1 s for backward crawling). This term ensures that PMN activity in the A1 and the A2 segments is similar but offset in time. The scaling term $\alpha_n$ increases quadratically from 0 to 0.1 over the 1000 training epochs. The final term $C_J = \alpha_n \left( ||\mathbf{J}^p - \mathbf{J}_0^p||^2 + ||\mathbf{J}^m - \mathbf{J}_0^m||^2 \right)$ penalizes deviations of model weights from the initial weights given by the TEM reconstruction. The correlation coefficient between the magnitudes of the nonzero entries of $\mathbf{J}^m$ and $\mathbf{J}_0^m$ was on average $0.87 \pm 0.01$ after optimization, and $0.43 \pm 0.02$ for $\mathbf{J}^p$ and $\mathbf{J}_0^p$, indicating that the optimization procedure found patterns of weights similar, though not identical, to those in reconstruction.

The total cost, equal to $C_{targ} + C_{A18b,A27h} + C_{seg} + C_J$, is optimized using the RMSProp optimizer for 1000 epochs. During each epoch, the costs corresponding to one FWD and one BWD trial are averaged. The learning rate decreases from $10^{-2}$ to $10^{-3}$ logarithmically over the course of optimization. Code is available at GitHub (*Mark and Litwin-Kumar, 2019*; copy archived at https://github.com/elifesciences-publications/larval_locomotion).

## Acknowledgements

We thank Luis Sullivan, Emily Sales, and Hiroshi Kohsaka for comments on early versions of the manuscript. BM was supported by an NIH training grant T32HD007348. AC was supported by HHMI. AL-K was supported by the Burroughs Wellcome Foundation, the Gatsby Charitable Foundation, the Simons Collaboration on the Global Brain, and NSF award DBI-1707398. CQD and AAZ were supported by HHMI and NIH HD27056.

## Additional information

### Funding

| Funder | Grant reference number | Author |
|---|---|---|
| Howard Hughes Medical Institute | | Aref Arzan Zarin<br>Albert Cardona<br>Chris Q Doe |
| National Institutes of Health | HD27056 | Brandon Mark<br>Chris Q Doe |
| Burroughs Wellcome Foundation | | Ashok Litwin-Kumar |
| Gatsby Charitable Foundation | | Ashok Litwin-Kumar |
| Simons Collaboration on the Global Brain | | Ashok Litwin-Kumar |
| NSF | DBI-1707398 | Ashok Litwin-Kumar |
| NIH | HD27056 | Chris Q Doe |

The funders had no role in study design, data collection and interpretation, or the decision to submit the work for publication.

### Author contributions

Aref Arzan Zarin, Data curation, Formal analysis, Validation, Investigation, Methodology; Brandon Mark, Conceptualization, Resources, Data curation, Software, Formal analysis, Validation, Investigation, Visualization, Methodology; Albert Cardona, Resources, Software, Funding acquisition, Methodology; Ashok Litwin-Kumar, Conceptualization, Resources, Software, Funding acquisition, Validation, Methodology; Chris Q Doe, Conceptualization, Data curation, Supervision, Funding acquisition, Investigation, Methodology, Project administration

### Author ORCIDs

Aref Arzan Zarin  https://orcid.org/0000-0003-0484-3622
Ashok Litwin-Kumar  http://orcid.org/0000-0003-2422-6576
Chris Q Doe  https://orcid.org/0000-0001-5980-8029

### Decision letter and Author response

Decision letter https://doi.org/10.7554/eLife.51781.sa1
Author response https://doi.org/10.7554/eLife.51781.sa2

## Additional files

### Supplementary files

• Supplementary file 1. CATMAID. json file of all reconstructed MNs in segment A1 as of 17 February 2019.

• Supplementary file 2. List of all PMN names and published synonyms.

• Supplementary file 3. Excel file of PMN neurotransmitter identity as of 31 August 2019. Data from our own work plus previous studies (*Kohsaka et al., 2014*; *Heckscher et al., 2015*; *Fushiki et al., 2016*; *Hasegawa et al., 2016*; *MacNamee et al., 2016*; *Zwart et al., 2016*; *Takagi et al., 2017*; *Yoshino et al., 2017*; *Burgos et al., 2018*; *Carreira-Rosario et al., 2018*; *Kohsaka et al., 2019*).

• Supplementary file 4. CATMAID. json file of all reconstructed PMNs in segment A1 as of 21 August 2019.

• Supplementary file 5. Excel file of PMN to MN connectivity as of 31 August 2019.

• Supplementary file 6. Excel file of PMN to PMN connectivity as of 31 August 2019.

• Transparent reporting form

### Data availability

All data generated or analyzed during this study are included in the manuscript and supporting files. Source code is available at https://github.com/alitwinkumar/larval_locomotion (copy archived at https://github.com/elifesciences-publications/larval_locomotion).

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
