## [Decision Letter]

**Acceptance summary:**

This work provides an extensive and comprehensive analysis of premotor interneurons (PMNs), motor neurons (MNs) and muscles governing *Drosophila* larval locomotion. The authors examine muscle recruitment during forward and backward crawling and identify specific muscles that show differential activation timing between forward and backward crawling. Examining the connectivity between MNs and PMNs with synaptic resolution reveals evidence for both a "labelled line" as well as "combinatorial code" type PMN-MN network. This represents a significant advance in our understanding of directional control of locomotion.

**Decision letter after peer review:**

Thank you for submitting your article "A multilayer circuit architecture for the generation of distinct locomotor behaviors in *Drosophila*" for consideration by *eLife*. Your article has been reviewed by three peer reviewers, and the evaluation has been overseen by a Reviewing Editor and Ronald Calabrese as the Senior Editor. The reviewers have opted to remain anonymous.

The reviewers have discussed the reviews with one another and the Reviewing Editor has drafted this decision to help you prepare a revised submission.

Summary:

This manuscript investigates the neural mechanisms by which two distinct aspects of locomotion, forward and backward movements, are brought about in *Drosophila* larva. It can be divided into three parts. The first part describes the muscles involved and their activity during forward and backward locomotion using calcium imaging; the second part utilizes the EM volume of the larval CNS to identify the motor neurons and premotor neurons, and their synaptic connectivity patterns; the third part invokes modeling to simulate how the basic network may function during the two behavioral events. The uniqueness of the paper lies in its comprehensiveness, as all (or nearly all) of the above components are evaluated at a whole segment level. Overall, this study represents a significant advance in our understanding of directional control of locomotion.

Essential revisions:

1) In the first experiment (Figure 1-3) the authors perform an elegant and comprehensive analysis of muscle activity during forward and backward crawling cycle. This culminates into identification of specific muscles (VO, 1, 2, 11) that are recruited at distinct times in forward versus backward crawling (Figure 3). This finding sets the stage for the rest of the paper for finding a model of how neurons recruit these muscles. Unfortunately, the following sections lack specific references (Except Figure 5D,E) to these muscles and the reader is left searching for an explanation in dense data figures and supp. Highlighting MNs and PMNs that recruit these specific muscles, in every section is essential for coherence, furthermore a lack of identification of MNs and PMNs to drive differential recruitment of some of these specific muscles, requires an explanation (e.g. VO muscle group).

2) In Figure 2-3, the authors suggest co-activation based hierarchical clustering as a better grouping of functional muscle types, compared to previously used "spatial clusters". However with respect to MN postsynaptic sites (Figure 5), a thorough examination (including hierarchical clustering analysis) is only performed for spatial clusters. Moreover co-activation based clusters show a weak segregation of postsynaptic MN sites. It would help if there were quantitative analysis of the clusters the authors have used for the remainder of the manuscript.

Additionally, authors could perform a comparison between MNs innervating differential recruited muscles and MNs innervating muscles that show identical recruitment during forward and backward crawling. This will inform whether the postsynaptic density could provide any useful information to explain the muscle recruitment.

3) The PMN-MN connectivity analysis is impressive in scope (number of neurons traced) and detail (neurotransmitter identities). Although the authors elegantly show potential for both "labelled line" and "combinatorial code" based on this connectivity, it would once more help if authors could highlight whether it is labelled line or combinatorial input that drives MNs for the specific muscles that show differential recruitment timing.

4) The authors show a first comprehensive connectivity constrained model of PMN-MN network capable of producing muscle activity patterns observed during crawling forwards and backwards. Although the model is neatly verified by accompanying imaging experiments, it would have been more satisfying if there were fewer constraints on PMN activity, given the authors are anyways optimizing the model to get the muscle activity observed in the imaging data. Specifically, the PMNs that show the most distinct recruitment during forward versus backward crawling (A27h and A18b3) are imposed in the model. It would be interesting to see if the differential A27h and A18b3 activation emerges from connectivity constraints. Furthermore, some speculation regarding how descending neurons might impose the activity patterns on the PMN-MN network will be beneficial.

---

## [Author Response]

Essential revisions:1) In the first experiment (Figure 1-3) the authors perform an elegant and comprehensive analysis of muscle activity during forward and backward crawling cycle. This culminates into identification of specific muscles (VO, 1, 2, 11) that are recruited at distinct times in forward versus backward crawling (Figure 3). This finding sets the stage for the rest of the paper for finding a model of how neurons recruit these muscles. Unfortunately, the following sections lack specific references (Except Figure 5D,E) to these muscles and the reader is left searching for an explanation in dense data figures and supp. Highlighting MNs and PMNs that recruit these specific muscles, in every section is essential for coherence, furthermore a lack of identification of MNs and PMNs to drive differential recruitment of some of these specific muscles, requires an explanation (e.g. VO muscle group).

This is an excellent point, with which we fully agree. We now modify Figures 3, 4, 5, 7, and 8 to better highlight the differentially active muscles, MNs and PMNs. In new Figure 3H-I, we add new data on the timing of the differentially-active muscles, and a larger summary as requested below. In new Figure 3—figure supplement 2, we identify the differentially-active muscles by blue/red coloring. In Figure 4, we label the differentially active MNs. In new Figure 5E-H, we characterize neuropil postsynaptic localization for the most differentially-active MNs (2, 11, 18, VO). In new Figure 7G, we have quantified the connectivity pattern between the 118 PMNs and the differentially-active MNs. Finally, in new Figure 8 we now show the cellular morphology of the differentially active MNs in new panels 8A-E. Thus, we now trace these differentially-active muscles and MNs throughout the paper, which gives a common thread for interpreting each dataset.

2) In Figure 2-3, the authors suggest co-activation based hierarchical clustering as a better grouping of functional muscle types, compared to previously used "spatial clusters". However with respect to MN postsynaptic sites (Figure 5), a thorough examination (including hierarchical clustering analysis) is only performed for spatial clusters. Moreover co-activation based clusters show a weak segregation of postsynaptic MN sites. It would help if there were quantitative analysis of the clusters the authors have used for the remainder of the manuscript.

We have reorganized the panels in Figure 5 and rewrote the result section to make it more reader friendly. We now clearly mention in the text that unbiased hierarchical clustering analysis results in better segregation of spatial clusters than the co-active muscle clusters (new Figure 5A). Also, we have now done a thorough examination of MN postsynaptic sites for forward (FWD) and backward (BWD) co-active clusters as well as for spatial clusters (new Figure 5B-D). We agree with your interpretation and now strengthen our conclusion that MNs innervating spatial muscle groups form a quantitatively better myotopic map compared to MNs innervating co-active muscle groups. See subsection “Co-active motor neurons have dispersed postsynaptic sites within the dorsal neuropil”.

Additionally, authors could perform a comparison between MNs innervating differential recruited muscles and MNs innervating muscles that show identical recruitment during forward and backward crawling. This will inform whether the postsynaptic density could provide any useful information to explain the muscle recruitment.

Following this helpful suggestion, we have made new panels (Figure 5E-H). We show that three out of four differentially recruited MNs show quantitatively different spatial segregation of postsynaptic sites when compared with MNs that are co-active with them during either forward and backward crawling. However, MN2 is differentially-recruited yet shows no clear segregation of postsynapses within the neuropil compared to other neurons in its FWD or BWD co-active muscle group. Thus, we conclude that distinct postsynaptic localization is not required to establish temporally distinct MN recruitment. See paragraph three of subsection “Co-active motor neurons have dispersed postsynaptic sites within the dorsal neuropil”.

3) The PMN-MN connectivity analysis is impressive in scope (number of neurons traced) and detail (neurotransmitter identities). Although the authors elegantly show potential for both "labelled line" and "combinatorial code" based on this connectivity, it would once more help if authors could highlight whether it is labelled line or combinatorial input that drives MNs for the specific muscles that show differential recruitment timing.

Thank you for this comment. We have now made a new Figure 7, where we address several questions revolving around "labelled line" vs "combinatorial code" models of connectivity. First, we show that while some PMNs show a “labelled line” connectivity patterns with MNs in a single forward or backward co-active muscle group, the majority of PMNs exhibit connectivity patterns consistent with the "combinatorial code" model (new Figure 7A-F). Next, in new Figure 7G, we show that essentially none of the PMNs specifically connect to individual MNs with differential recruitment during forward versus backward locomotion (MNs 2, 11, 18, and the two VO MNs). This also supports the "combinatorial code" model. See paragraph three of subsection “TEM reconstruction of 118 premotor neurons reveals premotor neuron pools targeting each group of co-active motor neurons”.

4) The authors show a first comprehensive connectivity constrained model of PMN-MN network capable of producing muscle activity patterns observed during crawling forwards and backwards. Although the model is neatly verified by accompanying imaging experiments, it would have been more satisfying if there were fewer constraints on PMN activity, given the authors are anyways optimizing the model to get the muscle activity observed in the imaging data. Specifically, the PMNs that show the most distinct recruitment during forward versus backward crawling (A27h and A18b3) are imposed in the model. It would be interesting to see if the differential A27h and A18b3 activation emerges from connectivity constraints. Furthermore, some speculation regarding how descending neurons might impose the activity patterns on the PMN-MN network will be beneficial.

We agree that inferring the selectivity of the A27h and A18b PMNs from the connectome alone would be compelling. We therefore re-ran our simulations but without a penalty to promote the activity of these neurons to be consistent with what is known. We found that the A27h neuron maintained its selectivity consistent with experiment, but A18b was no longer specifically active during backward locomotion. We interpret this to mean that the PMN-MN connectome is insufficient to capture the selectivity of A18b to backward locomotion, which is consistent with our recent study that showed that it is directly activated by a descending neuron not included in our model (Carreira-Rosario et al., 2018). In contrast, our model accurately predicts the FWD-specific activity of A27h, raising the possibility that A27h activity can be generated from the current pool of 118 PMNs without descending input. We have added a description of these results in paragraph two of subsection “A recurrent network model that generates the observed forward and backward pattern of muscle activity” and a new Figure 9—figure supplement 2.